# Toward Experiment-Guided Hypothesis Ranking via Simulated Experimental Feedback

## Abstract

Hypothesis ranking is a crucial component of automated scientific discovery, particularly in natural sciences where wet-lab experiments are *costly* and *throughput-limited*. Existing approaches focus on pre-experiment ranking, relying solely on a language model's internal reasoning without incorporating empirical outcomes. We introduce the task of *experiment-guided ranking*, which prioritizes hypotheses based on feedback from previously tested ones. However, developing such strategies in natural science domains is challenging due to the impractical requirement of repeatedly conducting real experiments. To address this, we revisit the core purpose of real experiments: to provide feedback on both the ground-truth hypothesis and the surrounding hypotheses that form the path toward it. This motivates our alternative: a simulator grounded in three domain-informed conceptual foundations, modeling hypothesis performance as a function of similarity to a known ground truth, perturbed by noise. While the ground-truth is pre-specified, it remains hidden from the ranking agent, enabling faithful evaluation of policies that navigate toward it. Validated against 124 hypotheses with experimentally reported outcomes, the simulator approximates real experimental results with consistent trend alignment. Though not perfectly accurate, its deviations resemble wet-lab noise and can foster more robust ranking strategies. We formulate experiment-guided ranking as a sequential decision-making problem and propose an in-context reinforcement learning (ICRL) framework. Within this framework, we introduce an LLM-based agentic policy that decomposes hypotheses into functional elements, clusters them by shared mechanistic roles, and prioritizes recombinations of promising elements based on feedback. Experiments show that our method significantly outperforms pre-experiment baselines and strong ablations. Our toolkit-comprising the simulator and ICRL framework—enables systematic research on experiment-guided ranking, with our policy serving as a strong proof of concept.

## 1 Introduction

Scientific discovery plays a foundational role in advancing human society (Coccia, 2019). Recent progress in large language models (LLMs) has sparked growing interest in automating parts of this scientific process (Luo et al., 2025). Among these, one of the most critical stages is hypothesis ranking: given a large set of automatically generated hypotheses (e.g., by AI), which one should be tested in a real experiment first? This question is particularly important in natural science domains, where wet-lab experiments are costly and throughput-limited, requiring prioritization strategies that maximize discovery efficiency under strict experimental budgets.

Existing work on hypothesis ranking (Yang et al., 2025; Si et al., 2024) primarily relies on evaluations based solely on a language model's internal reasoning, without incorporating any empirical feedback. We refer to this as *pre-experiment ranking*. While efficient, this approach overlooks the iterative, feedback-driven nature of real-world experimentation.

In contrast, we introduce the task of *experiment-guided ranking*, which prioritizes hypotheses for the next round of experimentation based on outcomes from previously tested ones. Rather than evaluating all candidates upfront, this approach dynamically adjusts prioritization as new experimental results become available. However, in natural science domains such as chemistry, materials science, and biology, conducting iterative experiments at scale—as required by experiment-

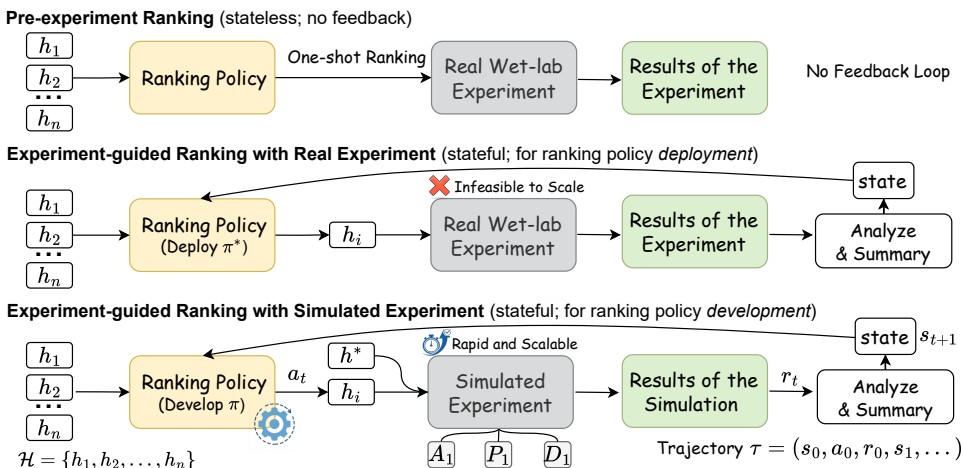

Figure 1: Overview of ranking strategies. Pre-experiment ranking is stateless and ignores feedback. Experiment-guided ranking with real experiments is stateful but infeasible to scale. Our simulator enables efficient testing of ranking policies through simulated feedback before real deployment.

guided ranking—is often infeasible due to the high cost, long duration, and limited throughput of real-world experimentation. This lack of scalable feedback limits progress in developing and evaluating experiment-guided ranking strategies.

To address this challenge, we revisit the core purpose of real experiments: not only to validate a ground-truth hypothesis, but also to provide feedback on nearby hypotheses that form the path toward it. This motivates our alternative: a simulator that approximates experimental feedback in a local neighborhood of hypothesis space, enabling the development and evaluation of experiment-guided ranking strategies.

Our simulator is grounded in three conceptual foundations, reflecting the universal natural-science principle that structural similarity implies similar behavior (Callister & Rethwisch, 1999; Hansch et al., 1995; Wiley, 1986; Alberts et al., 2015). *A1 (Local Optimum Assumption)* states that a ground-truth hypothesis represents a dominant local optimum within its sufficiently local neighborhood. *P1 (Scientific Principle)* holds that greater structural or functional similarity yields more similar outcomes. *D1 (Logical Deduction)* follows that, because similarity representations are imperfect, the observed performance landscape deviates from the ideal implied by A1 and P1.

We formalize these conceptual foundations and construct a simulator that models hypothesis performance as a function of distance to a hidden ground-truth hypothesis. Although the ground truth is known to the simulator, it remains hidden from the ranking policy—enabling rigorous evaluation of strategies that must infer it through limited feedback. To validate the simulator, we curate a dataset of 124 hypotheses with experimentally reported outcomes from the literature. Our simulator demonstrates high trend alignment and predictive accuracy in approximating real experimental outcomes. It also outperforms strong baselines adapted from prior work (Yang et al., 2025), further supporting its utility as a research tool for developing and evaluating experiment-guided ranking strategies. Though not perfectly accurate, its deviations resemble the noise observed in real wet-lab experiments and can foster more robust ranking strategies.

Building on this foundation, we develop an in-context reinforcement learning (ICRL) framework for experiment-guided hypothesis ranking. Within this framework, we instantiate a clustering-based agentic policy that decomposes hypotheses into functional components and groups them by shared mechanistic roles. After each experimental trial, the agent analyzes the tested hypothesis to infer which components contributed to its performance, then prioritizes untested hypotheses that incorporate the most promising functional elements. This enables efficient transfer of insights across structurally related candidates and helps navigate the hypothesis space more effectively. Experiments show that this policy significantly outperforms pre-experiment baselines and strong ablations. Combined with the simulator, our ICRL framework forms a general-purpose toolkit for studying experiment-guided ranking strategies, with our policy serving as a strong proof of concept. Figure 1 provides an overview

of the three paradigms: pre-experiment ranking, experiment-guided ranking with real experiments, and our simulator-driven approach for developing ranking policies.

Overall, the contributions of this paper are:

- We introduce and formalize the task of *experiment-guided ranking* and highlight a key bottleneck in the natural sciences: the lack of scalable access to wet-lab experimental feedback. To address this, we propose the use of simulators and release a curated dataset of 124 scientific hypotheses with annotated performance collected from the literature and real wet-lab experiments.
- We introduce three conceptual foundations characterizing the latent performance landscape of scientific hypotheses. We mathematically formalize this simulation process and construct a high-fidelity simulator that approximates real wet-lab outcomes by modeling performance as a function of hypothesis similarity and systematic distortion.
- We present a clustering-based agentic ranking policy implemented within an in-context reinforcement learning framework. It generalizes from limited feedback and outperforms both pre-experiment baselines and ablation variants.
- We provide a theoretical formalization of search complexity reduction via functional decomposition: demonstrating how attributing experimental feedback to the marginal utility of components allows for effective search space pruning. This transforms the discovery problem from an exponential combinatorial search into a linear component optimization, a theoretical result consistent with our empirical observations.

## 2 METHODOLOGY OF SIMULATOR CONSTRUCTION

### 2.1 CONCEPTUAL FOUNDATIONS AND FORMALIZATION

Our simulator construction is guided by three conceptual foundations—one assumption, one scientific principle, and one logical deduction—grounded in established principles of the natural sciences. Together, these provide a principled basis for modeling experimental outcomes of untested hypotheses, enabling systematic investigation of experiment-guided ranking strategies.

#### 2.1.1 CONCEPTUAL FOUNDATIONS

We posit that real experimental feedback within a hypothesis space can be simulated under the following conceptual foundations (*A1–P1–D1*):

1. (*A1: Local Optimum Assumption*) A ground-truth hypothesis represents a dominant local optimum within its sufficiently local neighborhood of the hypothesis space.
2. (*P1: Scientific Principle*) Hypotheses that are more similar in their underlying structure or function tend to yield more similar experimental outcomes.
3. (*D1: Logical Deduction*) In practice, representations of hypothesis similarity are imperfect proxies, so the resulting performance landscape deviates from the ideal implied by *A1* and *P1*, producing distortions such as noise, spurious local optima, or unexpected valleys.

*A1*, *P1*, and *D1* are all reasonable and sufficiently grounded. *P1* reflects the fundamental axiom that "structure determines properties, and properties determine outcome," which underpins multiple disciplines: molecular structure and material function in Chemistry & Materials Science (Callister & Rethwisch, 1999; Hansch et al., 1995), crystal structure and physical properties in Physics (Wiley, 1986), and protein structure and biological function in Biology (Alberts et al., 2015). *D1* follows logically from *A1* and *P1*, since any practical representation of hypothesis similarity must introduce distortions. *A1* is mostly valid but not guaranteed: even within a sufficiently small neighborhood, the labeled ground-truth hypothesis may not be the strict local optimum, as there could exist another hypothesis in that region with higher performance. This limitation, however, does not affect the simulator's role in developing ranking policies, whose goal is to recover the labeled ground truth. When deployed in real experiments, any superior hypotheses beyond the labeled ground truth would be directly revealed, ensuring that policies developed under a simulator supported by *A1* remain effective in practice.

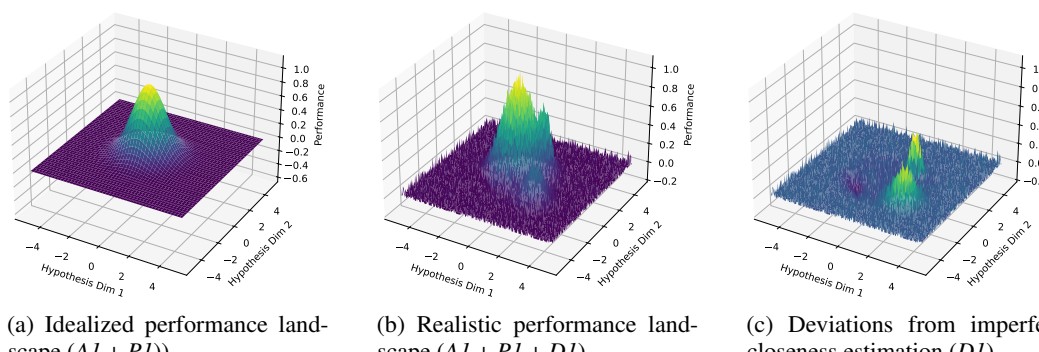

(a) Idealized performance landscape (*A1 + P1*)).

(b) Realistic performance landscape (*A1 + P1 + D1*).

(c) Deviations from imperfect closeness estimation (*D1*).

Figure 2: Illustration of the three conceptual foundations (*A1–P1–D1*) for simulator construction.

Figure 2 visually illustrates these conceptual foundations. In the ideal scenario (Figure 2a), *A1* ensures the presence of a dominant local optimum, while *P1* enforces that hypotheses closer in structure or function to this optimum yield more similar outcomes. Together, these yield a smooth, unimodal performance landscape, where Euclidean distance in hypothesis space faithfully reflects structural and functional similarity. However, practical scenarios differ substantially, as the measured distance ("closeness") between hypotheses—whether estimated by scientists or LLMs—may not faithfully capture structural and functional similarity. For instance, a chemical hypothesis might contain a useful functional component whose contribution is underrepresented, placing it farther from the dominant peak than warranted and creating a spurious secondary maximum. Conversely, a weaker hypothesis may appear deceptively close to the optimum, forming a local valley. These distortions yield a more irregular performance landscape, as illustrated in Figure 2b, with unexpected secondary peaks and valleys. Figure 2c further isolates these deviations, highlighting the gap between the idealized oracle landscape and practical estimates of closeness.

We now formalize these foundations by defining a mathematical model that makes explicit the relationship between hypothesis embeddings, similarity, and performance.

### 2.1.2 MATHEMATICAL FORMULATION

Let $\mathcal{H} \subset \mathbb{R}^d$ denote the hypothesis space, where each hypothesis $h \in \mathcal{H}$ is represented as a point in a $d$-dimensional latent space, conditioned on a specific research question $q$. Let $h^* \in \mathcal{H}$ denote the ground truth hypothesis for $q$, representing an experimentally validated optimum. We define the idealized performance function for any hypothesis $h$ in the vicinity of $h^*$ as:

$$f(h, h^*; q, \phi^*(\cdot)) = \frac{1}{(2\pi\sigma^2)^{d/2}} \exp\left(-\frac{\|\phi^*(h \mid q) - \phi^*(h^* \mid q)\|^2}{2\sigma^2}\right), \tag{1}$$

where $\phi^*(\cdot \mid q)$ is an oracle embedding function that maps each hypothesis $h$ to a point in the latent hypothesis space under the context of research question $q$. The embedded positions capture the oracle's understanding of closeness, measured by the Euclidean distance $\|\phi^*(h \mid q) - \phi^*(h^* \mid q)\|$.

We model the idealized performance surface as a Gaussian-like function centered at $\phi^*(h^* \mid q)$, yielding a strictly unimodal landscape that decays smoothly with increasing distance from the optimum $h^*$ (Figure 2a). While the true performance landscape in feature space may not be strictly Gaussian, the isotropic Gaussian form serves as a tractable and interpretable approximation in the latent space. This modeling choice directly reflects *A1* and *P1*.

However, practical simulations rely on imperfect embeddings of hypotheses into the latent space, stemming from limitations in domain understanding—no matter whether the embedding is performed (internally) by human experts or LLMs. Consequently, this leads to distortions in perceived "closeness", effectively warping the positions of hypotheses in latent space. Such a distorted hypothesis embedding $\tilde{\mathcal{H}}$ yields a different observed structure:

$$\tilde{f}(h, h^*; q, \phi(\cdot)) = f(h, h^*; q, \phi^*(\cdot)) + \epsilon(h \mid q) \tag{2}$$

where $\phi(\cdot \mid q)$ is a practical embedding function that maps each hypothesis $h$ into (somewhat distorted) positions in the latent hypothesis space for a research question $q$, and $\epsilon(h \mid q)$ represents

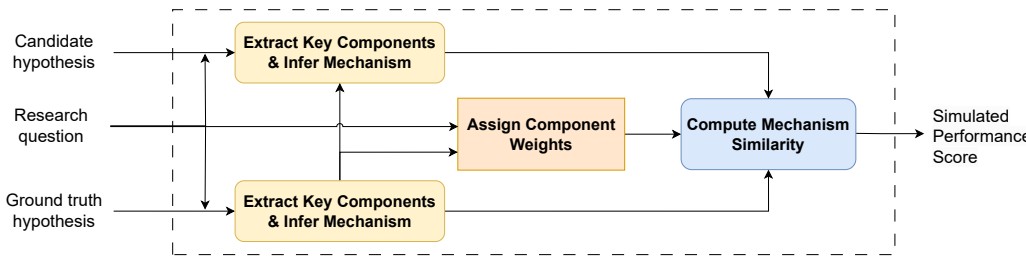

Figure 3: The internal structure of the simulator.

a systematic correction term that accounts for the discrepancy between oracle embedding $\phi^*(\cdot \mid q)$ and the practical embedding $\phi(\cdot \mid q)$ under the context of $q$. As a result, the practical embedding $\tilde{\mathcal{H}}$ introduces systematic distortions in the latent space, leading to spurious optima or valleys—effectively transforming the unimodal ideal surface into a noisier, multimodal one (Figure 2b).

Crucially, Figures 2a and 2b illustrate the same underlying performance-closeness relationship $f(h, h^*)$, differing only by $\phi(h)$, which is how hypotheses are embedded in the latent space. Figure 2c illustrates $\epsilon(h)$, the correction term that accounts for the discrepancy between the oracle embedding $\phi^*(\cdot)$ and the practical embedding $\phi(\cdot)$.

## 2.2 A Practical Implementation of $\phi(\cdot)$ with Prior Knowledge

As discussed in § 2.1, the core objective of the simulator is to construct an embedding function $\phi(\cdot)$ that maps each hypothesis $h$ into a latent space such that distances in this space reflect meaningful functional differences. Through extensive discussions with domain experts, we observe that a scientific hypothesis succeeds in addressing a research question primarily due to its underlying mechanisms.

Specifically, an effective hypothesis typically comprises a set of scientifically meaningful components—each contributing to distinct yet complementary sub-mechanisms—which together enable the overall reaction to fulfill its intended function. The specific prompts and examples for extracting key components and inferring mechanisms are provided in § A.

Informed by this domain knowledge, we design a simulator architecture illustrated in Figure 3. Each module corresponds to a subroutine implemented using an LLM with task-specific prompting. The simulator's goal is to estimate the latent-space distance $\|\phi(h \mid q) - \phi(h^* \mid q)\|$ between a candidate hypothesis $h$ and a ground truth hypothesis $h^*$, conditioned on a research question $q$.

The simulation begins by decomposing both the candidate and ground truth hypotheses into a set of key functional components, and identifying the underlying mechanism associated with each component in the context of the research question. The decomposition of $h^*$ is performed first, serving as a reference. These reference components and mechanisms guide the decomposition of $h$, ensuring alignment in both granularity and mechanistic interpretation.

Concurrently, the *Assign Component Weights* module estimates the relative importance $w_i$ of each component in the ground truth hypothesis, given the research question. A subset of these components—denoted $\mathcal{C}$—are labeled as critical, meaning they are considered necessary for the reaction to succeed. To elaborate on the role of $\mathcal{C}$, we provide illustrative examples in § B.

Next, the *Compute Mechanism Similarity* module compares each key component in $h^*$ with its corresponding component in $h$, assigning a similarity score $s_i \in [0, 1]$ to each pair. These scores are then aggregated using a weighted sum, combined with a multiplicative penalty that enforces the presence of all critical components:

$$S(h \mid q; h^*) = (\prod_{i \in \mathcal{C}} \mathbf{1}_{s_i > 0}) \cdot (\sum_{i=1}^{K} w_i \cdot s_i), \quad \text{where} \quad \sum_{i=1}^{K} w_i = 1 \tag{3}$$

This formulation guarantees that $S(h^* \mid q; h^*) = 1$, since all components are present with maximal similarity ($s_i = 1$ for all $i$), resulting in zero distance from the ground truth. Similarity score $S$ are

Figure 4: Experiment-guided ranking policy within an in-context reinforcement learning framework.

thereby bounded in $[0, 1]$, and lower distances correspond to stronger functional alignment with the ground truth hypothesis. The resulting value is used as the simulated performance score.

The final distance between the candidate and ground truth hypotheses is then calculated as:

$$|\phi(h \mid q) - \phi(h^* \mid q)| = |S(h \mid q; h^*) - 1| \tag{4}$$

## 3 METHODOLOGY OF EXPERIMENT-GUIDED RANKING

### 3.1 TASK FORMULATION

Given a research question $q$, a set of candidate hypotheses $\mathcal{H}$ is formed by selecting hypotheses generated by existing scientific discovery systems (Yang et al., 2025) and ground-truth hypotheses from top-tier scientific journals reporting high-quality lab experiments. The goal of experiment-guided ranking is to identify the optimal hypothesis $h^* \in \mathcal{H}$ with the highest experimental performance using an experiment executor $E$. Formally, we define the experiment executor as a function:

$$E : \mathcal{H} \to [0, 1] \tag{5}$$

that maps each hypothesis $h \in \mathcal{H}$ to a normalized performance score $s \in [0, 1]$. The normalization provides a unified performance metric across heterogeneous hypotheses and varying problem settings $q$, and can be defined relative to a domain-specific state-of-the-art benchmark established by experts.

The primary goal is to find $h^*$. However, since each evaluation of $E(h)$ corresponds to a real or simulated experiment—which may be costly or time-consuming—a critical requirement is to identify $h^*$ using as few experimental trials as possible. Accordingly, an effective experiment-guided ranking strategy must actively incorporate feedback from prior evaluations to guide subsequent selections, balancing exploration and exploitation under a limited experimental budget.

Thus, the problem can be reframed as finding a selection strategy that minimizes the number of trials required to identify the optimal hypothesis:

$$\arg\min_{\pi} \ N_{\text{trials}}^{\pi} \quad \text{subject to} \quad h^* = \arg\max_{h \in \mathcal{H}} E(h), \tag{6}$$

where $\pi$ denotes the hypothesis selection strategy, and $N_{\text{trials}}^{\pi}$ is the number of experiments required under strategy $\pi$ to successfully discover $h^*$.

### 3.2 METHODOLOGY

Due to the high cost and data-scarce nature of wet-lab experiments in the natural sciences, conventional reinforcement learning (RL), which relies on extensive interaction and parameter updates, is often impractical. Our approach circumvents this bottleneck by formulating the learning process within the context window of a frozen large language model. This gradient-free, non-parametric paradigm relies solely on forward passes, enabling the agent to learn from minimal trials without costly fine-tuning. The framework leverages the LLM's intrinsic reasoning capabilities, ensuring excellent generalizability to diverse scientific discovery tasks. Our agent, CSX-Rank, learns an optimal hypothesis selection policy via a formal sequential decision-making process.

We formulate experiment-guided ranking as a sequential decision-making process. At each timestep $t$, the agent observes a state $s_t$ representing the cumulative analysis of past experiments. It then performs an action $a_t$ by selecting a hypothesis $h \in H$ to test, receiving a reward $r_t$ from the experimental outcome. The trajectory is thus $\tau = [s_0, a_0, r_0, \ldots]$.

Unlike standard RL settings that maximize cumulative reward, our objective is to identify the optimal hypothesis using the minimum number of experiments, reflecting the high cost of scientific exploration. The agent's goal is to learn an optimal policy $\pi^*$ that minimizes the expected trials:

$$\pi^* = \arg \min_{\pi} E_{\pi}[N_{\text{trials}}] \tag{7}$$

Here, an effective policy $\pi(s_t)$ leverages the accumulated knowledge in the state to make more strategic selections, thus minimizing $N_{\text{trials}}$. Our agent, CSX-Rank, implements this policy through the structured, iterative process detailed below (Figure 4).

**Step 1: Extraction, Classification, and Clustering of Functional Components.** To generalize from specific results, the agent decomposes each hypothesis $h \in H$ into functional components, which are classified as effective, uncertain, or ineffective; the latter are pruned for efficiency. The remaining components are clustered by functional similarity, with each cluster representing a distinct mechanistic contribution to solving $q$. This yields a structured state representation $s_t$, where each element remains traceable to its originating hypothesis.

**Step 2: Cluster and Hypothesis Selection.** To connect abstract mechanistic knowledge (clusters) with a concrete experiment (hypothesis), the policy $\pi(s_t)$ selects the next action $a_t$ through a two-stage process. First, guided by prior domain knowledge, the LLM identifies the most promising cluster. Within this cluster, it then selects the most relevant hypothesis $h$, which defines the action $a_t$.

**Step 3: Experiment Execution and Result Analysis.** The selected hypothesis $h$ (action $a_t$) is evaluated by the executor $E$—either our high-fidelity simulator (CSX-Sim) or a real wet lab—which returns a normalized performance score $s \in [0, 1]$. This score serves as the reward $r_t = E(a_t)$, and its analysis quantifies the action's success, grounding the policy in empirical results.

**Step 4: Iterative Summarization and Refinement.** To make learning cumulative, the agent integrates each experimental outcome into a running summary. This updated summary forms the new state $s_{t+1}$ for the next decision cycle, closing the RL loop and enabling the policy to refine systematically from prior knowledge and new feedback.

A key strength of our multi-step, component-driven framework is its inherent interpretability. By design, the agent must break down its decision process into an explicit, auditable trail—from extracting and clustering components to selecting the final hypothesis. Such structured and transparent reasoning is essential in scientific applications, allowing domain experts to examine the agent's logic, build trust in its recommendations, and derive new insights.

## 4 EXPERIMENT

We name our simulator as *CSX-Sim*, and the experiment-guided ranking method as *CSX-Rank*. All experiments are implemented with `GPT-4o-mini` (OpenAI, 2024).

### 4.1 SIMULATOR: EVALUATING THE SIMULATOR WITH REAL EXPERIMENT RESULTS

| Simulator | Spearman Correlation (↑) | Perfect Consistency Indicator (↑) | RMSE (↓) |
|---|---|---|---|
| Matched Score | 0.843 | 12/30 | 0.232 |
| *CSX-Sim* | **0.960** | **26/30** | **0.213** |
|   w/o CriticalPoints | 0.950 | 23/30 | 0.229 |
|   w/o ComponentExtraction | 0.864 | 12/30 | 0.272 |

Table 1: Simulator validation against real-world wet-lab results.

We curated a benchmark of 30 research questions and 124 hypotheses from published literature, each with experimentally validated outcomes spanning multiple domains (§ C.1). For each hypothesis, simulated results from *CSX-Sim* were compared against the annotated outcomes (§ C.2). Evaluation considered two criteria: (1) *Trend alignment*, measured by Spearman correlation, assessing whether predicted performances preserve the relative ordering of ground-truth outcomes. Because ranking depends on relative differences, we also report the Perfect Consistency Indicator (PCI), the number of questions with perfect alignment. (2) *Predictive accuracy*, measured by RMSE, capturing absolute deviations between predicted and experimental values (see § D for details and additional indicators). Comparative results are shown in Table 1.

**Baseline and Ablation**   We adopt the "Matched Score" (Yang et al., 2025) as our primary baseline, which evaluates hypotheses by measuring their similarity to ground-truth references through a reference-based comparison. Additionally, we conduct two ablation studies on *CSX-Sim* to assess the contribution of its key components: (1) The first ablation (w/o CriticalPoints) disables the labeling of critical components $\mathcal{C}$, as defined in Equation 3, allowing hypotheses that lack essential components to still receive positive feedback from the simulator; (2) The second ablation (w/o ComponentExtraction) skips the extraction and weighting of critical components, directly computing mechanism similarity using prompts analogous to the final module in Figure 3.

**Results Interpretation**   As shown in Table 1, *CSX-Sim* outperforms baselines across all metrics, demonstrating stronger trend alignment, greater robustness, and lower predictive error. Compared to the Matched Score baseline, it achieves notable gains in correlation and consistency while reducing error. Ablation studies confirm the importance of component analysis: removing CriticalPoints causes modest degradation, whereas omitting component extraction leads to substantial drops in alignment and accuracy. These results highlight the necessity of fine-grained component analysis for high-fidelity simulation feedback.

## 4.2 EXPERIMENT-GUIDED RANKING: BASELINES AND ABLATION STUDY

**Data and Evaluation Metrics**   We evaluate experiment-guided ranking on the TOMATO-chem dataset (Yang et al., 2025), which contains 51 scientific problems, each annotated with a ground-truth hypothesis. For each problem, the MOOSE-Chem framework (Yang et al., 2025) generates 63 additional candidates distinct from the ground truth, yielding 64 hypotheses per question (1 ground truth and 63 negatives). The disciplinary distribution is provided in § C.3. The dataset's interdisciplinary nature, evident in its inclusion of topics from fields such as applied physics and biology, stems from its origin in scientific literature where "chemistry" papers are frequently co-labeled with other scientific fields. Performance is measured by $N_{\text{trials}}$, the number of simulation-based evaluations needed to identify the ground-truth hypothesis for each problem. Lower $N_{\text{trials}}$ indicates more efficient prioritization. Results appear in Table 2.

| Method | $N_{\text{trials}}$ ($\downarrow$) |
|---|---|
| Uninformed Search | 32.500 |
| Pre-Experiment Ranking | 28.608 |
| *CSX-Rank* | **15.196** |
|    w/o Clustering | 27.980 |
|    w/o Clustering & Analysis | 35.627 |
|    w/o Clustering & Analysis & Full Feedback | 37.667 |

Table 2: Number of experiments required to identify the ground truth hypothesis across methods.

**Baselines**   We compare against two strategies: *Uninformed Search* and *Pre-Experiment Ranking*. Uninformed search selects hypotheses uniformly at random; Pre-experiment ranking scores hypotheses using only prior model knowledge, without feedback (Yang et al., 2025). As shown in Table 2, Uninformed Search require over 32 trials on average, while Pre-Experiment Ranking reduces this to under 30—outperforming both naive baselines but still far behind CSX-Rank. This indicates that relying solely on prior knowledge yields only modest gains without feedback, whereas experiment-guided ranking substantially improves sample efficiency. Detailed scalability analysis is in § I.

**Ablation Study**   To assess the contribution of key components in *CSX-Rank*, we conducted ablation studies under three conditions: (1) removing functional clustering (*CSX-Rank w/o Clustering*); (2) further disabling feedback analysis (*CSX-Rank w/o Clustering & Feedback Analysis*); and (3) additionally limiting feedback to the 10 most recent simulation results (*CSX-Rank w/o Clustering & Feedback Analysis & Full Feedback*). As shown in Table 2, progressively removing these components leads to marked performance degradation, confirming the importance of clustering, analytical summarization, and sufficient feedback quantity for efficient hypothesis ranking.

### 4.3 THEORETICAL COMPLEXITY REDUCTION.

We provide a theoretical formalization of search complexity reduction via functional decomposition. Formally, let $\mathcal{K}$ denote the universal discrete set of functional component clusters, and define the hypothesis space as $\mathcal{H} = \{h \mid h = \{k_1, \ldots, k_n\} \subseteq \mathcal{K}, |h| \leq m\}$, where $m$ bounds the maximum structural complexity of a hypothesis. Traditional strategies rely on holistic evaluation to optimize the joint probability $P(h)$, thus suffering from the curse of dimensionality with a search complexity scaling as $\mathcal{O}(|\mathcal{K}|^m)$. In contrast, CSX-Rank factorizes the optimization domain by attributing experimental feedback $y$ to the marginal utility of individual modules $k \in \mathcal{K}$. By updating the module-level posterior $P(k|y)$, our framework simultaneously adjusts the likelihood for the entire subset of candidates $\{h' \in \mathcal{H} \mid k \in h'\}$. This effectively acts as a massive pruning operator: a negative feedback on a single module allows the agent to discard a significant fraction of the total combinatorial space without individual testing. Consequently, the asymptotic complexity is theoretically reduced from exponential (in the combinatorial space) to linear (relative to the functional clusters, $\mathcal{O}(|\mathcal{K}|)$), decoupling the discovery cost from the hypothesis length $m$. We provide the rigorous mathematical derivation and formal proof in § M.

### 4.4 SIMULATOR: ABLATION ON DIFFERENT $\phi(\cdot)$ WITH DIFFERENT LEVELS OF DISTORTION

To study how simulator fidelity affects ranking, we note that experiment-guided ranking is essentially an optimization process over hypothesis space. A high-fidelity simulator provides informative feedback to guide this search, while distortions mislead it. We therefore introduce controlled distortions into $\phi(\cdot)$ to simulate increasingly challenging feedback conditions.

In collaboration with domain experts, we designed three distortion types commonly observed in practice—local maxima/minima, plateaus, and cliffs—reflecting typical challenges in hypothesis evaluation. We further defined three distortion levels (Simple, Moderate, Complex), incorporating progressively more noise into $\phi(\cdot)$; full details appear in § E.

We evaluated *CSX-Rank*, *CSX-Rank w/o Clustering*, and *CSX-Rank w/o Clustering & Analysis* across these noise conditions. As shown in Table 3, higher noise complexity consistently degraded performance, increasing $N_{\text{trials}}$. Still, *CSX-Rank* outperformed its ablated variants, preserving a clear efficiency margin even under Complex Noise (32.7 vs. 36.5 and 40.5 trials). These results demonstrate the robustness of clustering and feedback analysis in mitigating misleading signals and maintaining search efficiency, aligning with Section 4.2.

| Method | $N_{\text{trials}}$ (Simple Noise) | $N_{\text{trials}}$ (Medium Noise) | $N_{\text{trials}}$ (Complex Noise) |
|---|---|---|---|
| *CSX-Rank* | 21.804 | 26.608 | 32.706 |
| w/o Clustering | 32.706 | 35.843 | 36.471 |
| w/o Clustering & Analysis | 37.235 | 38.373 | 40.451 |

Table 3: Simulator with different noise conditions

## 5 CONCLUSION

We introduced the task of *experiment-guided ranking* and addressed its central bottleneck—the lack of scalable experimental feedback—by proposing a simulator grounded in three domain-informed conceptual foundations. Validated against 124 hypotheses, the simulator enables systematic evaluation of ranking policies. Building on this, we developed an in-context reinforcement learning framework with a clustering-based agentic policy that significantly outperforms pre-experiment baselines. Together, the simulator and policy provide a toolkit for advancing feedback-driven hypothesis discovery, with potential impact across the natural sciences.

## ETHICS STATEMENT

This work aims to accelerate beneficial scientific discovery, and the authors have read and adhered to the ICLR Code of Ethics. We acknowledge the potential for dual-use applications inherent in this research, as well as potential biases from our literature-based datasets and simulator conceptual foundations. We believe the benefits of a transparent and auditable framework for research outweigh these risks and are committed to its responsible application. No human subjects were involved in this study.

## REPRODUCIBILITY STATEMENT

All code and data supporting this research are publicly available in an anonymous repository, a link to which is provided on the first page of this paper.

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

## A    Extracting Key FUNCTIONAL Components in the Simulator

### A.1    A Framework for Extracting Critical Functional Components in the Simulator

The specific framework of *CSX-Sim* for extracting key functional components is illustrated in Figure 5. To demonstrate this process, we categorize the critical components and conclusions embedded within scientific hypotheses using a representative example from the field of chemistry. We then analyze the specific role and underlying mechanism of each component in the context of the research problem and the derived conclusions. Finally, the system synthesizes and extracts the key components, their corresponding mechanisms, and the validated conclusions from the hypothesis.

**Framework for Analyzing Scientific Hypotheses in Chemical Problems**

**1. Identify Key Chemical Components and Conclusions**

$$H_i = \{K_i, C_i\}$$

($H_i$: Hypothesis $i$, $K_i$: Key Chemical Components, $C_i$: Conclusion)

**2. Explain Mechanism of Key Chemical Components**

$$M_i = f(K_i, S, C_i)$$

($M_i$: Mechanism, $S$: Scientific Problem, $K_i$, $C_i$ as above)

**3. Verify and Output Key Points, Mechanisms, and Conclusions**

$$O_i = \{P_i, M_i, C_i\}$$

($O_i$: Output, $P_i$: Key Points incl. $K_i$, $M_i$, $C_i$ as above)

Figure 5: A Framework for Extracting Chemical Components in the Simulator.

### A.2    Prompt for Extracting Key Chemical Components in the Simulator

The prompt for extracting key chemical components in the simulator, along with examples, is as follows:

*You are an experienced expert. I will provide you with a scientific question and a scientific hypothesis. Your task is to identify the chemical key points within the hypothesis that are essential for addressing the scientific question. Chemical key points are the core elements—such as basic chemical components, reactions, or mechanistic methods—critical to solving the problem effectively. Analyze these key points by linking them to the scientific question, determining how they contribute to resolving it.*

*When identifying chemical key points, consider the following:*

*Each substance may be a key point. If it includes specific parameters like concentration or mass fraction (e.g., 0.3M NaCl, 10wt% PVA), ensure these details are retained in the division process without losing specificity. If multiple substances are related and function together (e.g., potassium ferricyanide and potassium ferrocyanide as an oxidizing-reducing pair), group them as a single chemical key point based on their shared role or interdependence. Exclude elements from the scientific question that reappear in the hypothesis as prerequisites (e.g., if the question involves improving MXene nanosheets and the hypothesis enhances them with liquid metal, MXene nanosheets are a prerequisite, not a key point; liquid metal is the key point). Prerequisites should not be output or analyzed as key points. Distinguish key points from validation methods (e.g., elemental analysis to verify properties). Validation methods support the hypothesis but are not chemical key points. For each identified chemical key point, conduct a detailed and rigorous analysis of its role and function in*

*relation to the scientific question. Use your chemical knowledge to explain the specific mechanism by which it addresses the problem, focusing on how it enhances the relevant properties or performance outlined in the question. Provide a clear, mechanistic explanation of its contribution and, if multiple key points exist, describe their interconnections.*

*Additionally, identify the results—effects or phenomena caused by these key points—representing the experiment's outcomes. In your output, focus on listing and explaining the chemical key points, followed by the results, ensuring no prerequisites from the scientific question are included.*

*Output format:*

*Chemical Key Points Chemical substance/component/method 1*
*Role and Function: Describe the role and function of the substance or method, including a detailed mechanistic explanation of how it addresses the scientific question and enhances relevant properties.*
*Chemical substance/component/method 2*
*Role and Function: Describe the role and function of the substance or method, including a detailed mechanistic explanation of how it addresses the scientific question and enhances relevant properties.*
*End Chemical Key Points Results Result 1:*
*Describe the effects caused by the aforementioned reasons (e.g., performance improvement, efficiency changes).*
*Result 2:*
*Further describe other effects related to the experimental objectives.*
*End Results*

***Example:*** *Chemical Key Points 1. 10wt% PVA (Polyvinyl Alcohol)*
*Role and Function: Polyvinyl alcohol (PVA) hydrogel acts as the base material, providing structural support and mechanical performance for thermoelectric gels. PVA with a mass fraction of 10% can provide mechanical support through hydrogen bonds in its structure and interact with potassium ferricyanide and potassium ferrocyanide to offer electrical changes.*
*2. $Gdm_2SO_4$ (Guanidine Sulfate)*
*Role and Function: Guanidine sulfate ($Gdm_2SO_4$) is integrated into the $K_3[Fe(CN)_6]$ / $K_4[Fe(CN)_6]$ to improve thermoelectric performance. The introduction of guanidine salt increases solvent entropy and effectively enhances thermopower.*
*3. Directional Freezing Method*
*Role and Function: By employing directional freezing technology, aligned channels are created, enhancing the electrical conductivity and mechanical strength of the material.*
*4. Potassium Ferricyanide and Potassium Ferrocyanide ($K_3[Fe(CN)_6]$ / $K_4[Fe(CN)_6]$)*
*Role and Function: These compounds are crucial electrolytes that facilitate redox reactions within the polymer gel. The presence of these ions enhances ion mobility and conductivity due to their ability to undergo reversible redox processes, thereby boosting the thermoelectric properties of the gel*
*End Chemical Key Points Results Carnot-relative Efficiency*
*The Carnot-relative efficiency of the FTGA exceeds 8%.*
*Thermopower and Mechanical Robustness*
*Thermopower and mechanical robustness are enhanced, outperforming traditional quasi-solid-state thermoelectric cells.*
*End Results*

To better illustrate the effectiveness of extracting key components, we provide a detailed example in materials science where we compare the performance of our simulator against human experts on a real-world problem.

- Scientific Question: How can a cost-effective N-type quasi-solid-state thermocell be developed to boost electricity production from low-grade heat by improving both **ion transport** efficiency and **electrode** performance?

- Scientific Hypothesis:Develop a flexible N-type quasi-solid-state thermocell by integrating **anisotropic** polymer networks and **hierarchical 3D copper electrodes** to enhance ion transport, mechanical robustness, and thermoelectric performance. Utilizing Polyvinyl Alcohol (PVA) as the hydrogel matrix, the anisotropic structure is achieved through a directional freeze-thawing (DFT) process, which involves applying a temperature gradient during freezing to guide ice crystal growth for polymer chain alignment. Repeated cycles further enhance the alignment and crosslinking, creating anisotropic pores that reduce

ion transport resistance. Ionic crosslinking with a 0.7 M $CuSO_4$ electrolyte and 0.1 M $H_2SO_4$ strengthens the hydrogel while retaining flexibility. Meanwhile, hierarchical 3D copper electrodes, fabricated via oxidation, etching, and thermal reduction, provide a high surface area, enhancing redox kinetics of the $Cu^{2+}/Cu^0$ couple and obviating platinum electrode reliance. This synergistic design achieves a remarkable 1500% increase in power density, reaching 0.51 mW m$^{-2}$ at a $\Delta T$ of 5°C, with a thermopower of 0.7 mV K$^{-1}$ and ionic conductivity improved by 20%. Mechanical tests reveal significant strength with a tensile strain at break of 350% and the system maintains stability under various mechanical deformations. This approach offers a cost-effective, adaptable solution for low-grade heat harvesting.

Here's Expert Extracting Key Component:

- Key Points:
    1. Anisotropic Polymer Networks
       Role and Function: The layered polymer network structure enhances ion transport rates, thereby improving thermoelectric conversion efficiency.
    2. Hierarchical 3D Copper Electrodes
       Role and Function: The use of 3D copper electrodes effectively increases the reaction contact area, enhancing the thermoelectric potential.
    3. Polyvinyl Alcohol (PVA) as the Hydrogel Matrix
       Role and Function: PVA hydrogel is used as the matrix to provide good mechanical properties and structural support.
    4. Directional Freeze-Thawing (DFT) Process
       Role and Function: The DFT process forms ion channels along the freezing direction, thereby increasing ion transport rates.
    5. 0.7 M $CuSO_4$ Electrolyte
       Role and Function: This electrolyte is used for ionic crosslinking to enhance the mechanical properties of the thermoelectric material.
    6. 0.1 M $H_2SO_4$
       Role and Function: This acid is used together with a 0.7 M $CuSO_4$ electrolyte for ionic crosslinking to enhance the mechanical properties of the thermoelectric material.

- Results:
    - Enhanced Electrical Properties of the Thermoelectric Material
      - Increased power density
      - Improved thermopower
      - Enhanced ionic conductivity

    - Improved Mechanical Properties of the Thermoelectric Material
      - Increased strength with a tensile strain
      - Enhanced stability under various mechanical deformations

Here's *CSX-Sim* Extracting Key Component:

- Key Points:
    1. Anisotropic Polymer Networks
       Role and Function: These networks are engineered to enhance ion transport efficiency while providing mechanical robustness through alignment of polymer chains, reducing ion transport resistance in thermocells.
    2. Hierarchical 3D Copper Electrodes
       Role and Function: The use of hierarchical 3D copper electrodes increases the surface area, enhancing redox kinetics of the $Cu^{2+}/Cu^0$ couple, thus improving the thermoelectric performance without relying on expensive platinum electrodes.
    3. Copper Sulfate ($CuSO_4$) Electrolyte
       Role and Function: A 0.7 M $CuSO_4$ solution serves as an ionic crosslinker in the hydrogel, enhancing mechanical strength while facilitating efficient ion transport in the system.

4. Sulfuric Acid ($H_2SO_4$)
   Role and Function: The addition of 0.1 M $H_2SO_4$ in the hydrogel contributes to ionic crosslinking and enhancing the overall conductivity of the electrochemical system.

5. Directional Freeze-Thawing (DFT) Process
   Role and Function: This method is used to create an ordered structure in the polymer hydrogel, promoting alignment of polymer chains and ensuring anisotropic pores that further reduce ion transport resistance.

- Results:
  - Power Density Increase
  - Enhanced Thermopower
  - Improved Ionic Conductivity
  - Mechanical Strength under Deformation

Here's a comparison of the analysis results between our simulator and human experts:

By comparing the approaches of an expert and *CSX-Sim* in extracting key scientific components for the specific scientific issues of ion transport efficiency and electrode performance, *CSX-Sim* successfully identifies solutions in its scientific hypotheses, including anisotropic polymer networks and hierarchical 3D copper electrodes. Compared to the human expert, *CSX-Sim* captures five out of six key points, missing only one: "Polyvinyl Alcohol (PVA) as the Hydrogel Matrix." The points it does identify align accurately with those proposed by the human expert based on the hypothesis, demonstrating the high accuracy of *CSX-Sim* in extracting key scientific components.

## B    THE ROLE OF CRITICALPOINTS IN *CSX-Sim*

To better illustrate the role of labeling critical components $\mathcal{C}$ in *CSX-Sim*, as defined in Equation 3, we provide an example for clarity. For simplicity, we define the term $\left(\prod_{i \in \mathcal{C}} \mathbf{1}_{s_i > 0}\right)$ from Equation 3, related to CriticalPoints, as the *Correction Factor*. This factor takes values of either 0 or 1.

The scientific problem under study is: *How can a polymer gel material be designed to enhance the Seebeck coefficient (Se) by optimizing the matrix material and redox pair, thereby improving the energy conversion efficiency of a thermoelectric device utilizing the temperature difference between body heat and the environment?*

This scientific problem corresponds to four real experimental hypotheses, outlined as follows:

1. **Hypothesis 1**: By combining gelatin with KCl, prepare a gel with high ionic conductivity to investigate its Seebeck coefficient (Se) performance with the $[Fe(CN)_6]^{3-}/[Fe(CN)_6]^{4-}$ redox pair. KCl, as an electrolyte, significantly enhances the gel's ionic conductivity, while the $[Fe(CN)_6]^{3-}/[Fe(CN)_6]^{4-}$ redox pair boosts the Seebeck coefficient through temperature-gradient-driven ion diffusion. Gelatin provides biocompatibility and mechanical strength, making it suitable for efficient thermoelectric energy conversion.

2. **Hypothesis 2**: By combining a PVA matrix with HCl, prepare a gel with high ionic conductivity and investigate its Seebeck coefficient (Se) performance under the influence of the $Fe^{3+}/Fe^{2+}$ redox pair. HCl, as a strong electrolyte, significantly enhances the gel's ionic conductivity, while the $Fe^{3+}/Fe^{2+}$ redox pair boosts the Seebeck coefficient through temperature-difference-driven ion diffusion. PVA provides flexibility and transparency, and by optimizing the HCl concentration and PVA crosslinking degree, ion migration efficiency can be further improved, enhancing the Seebeck coefficient and making it suitable for efficient energy conversion in body-heat thermoelectric devices.

3. **Hypothesis 3**: By preparing a pure PVA gel, investigate its Seebeck coefficient (Se) performance under the influence of the $Fe^{3+}/Fe^{2+}$ redox pair. PVA, as a hydrophilic polymer, possesses a certain level of ionic conductivity, and the $Fe^{3+}/Fe^{2+}$ redox pair generates a Seebeck coefficient through temperature-difference-driven ion diffusion.

4. **Hypothesis 4**: By polymerizing acrylamide (PAM) to prepare a hydrogel and investigate its thermoelectric performance. The porous network structure of the polyacrylamide hydrogel enhances the gel's ionic conductivity.

The results of the actual wet-lab experiments are summarized in Table 4. For the first three hypotheses, the CriticalPoints are included in the assumptions, resulting in a Correction Factor of 1 and non-zero final results. In contrast, for the fourth hypothesis, the CriticalPoints are not included in the assumption, leading to a Correction Factor of 0 and a final result of 0.

Table 4: The Seebeck coefficient (Se) of polymer thermoelectric materials

| Matrix | The Seebeck coefficient Se / mV K$^{-1}$ |
|---|---|
| $[Fe(CN)_6]^{3-}$/$[Fe(CN)_6]^{4-}$ KCl/Gelatin | 17 |
| $Fe^{3+}$/ $Fe^{2+}$ HCl/PVA | 1.02 |
| $Fe^{3+}$/ $Fe^{3+}$ PVA | 0.85 |
| Polymerizing Acrylamide (PAM) | 0 |

Our simulator analyzes Hypothesis 4, where the Correction Factor is 0, as follows:

***Analysis:*** *The Ground Truth Scientific Hypothesis identifies three core key points: Gelatin, KCl (Potassium Chloride), and the $[Fe(CN)_6]^{3-}$/$[Fe(CN)_6]^{4-}$ Redox Pair, each playing significant roles in enhancing the Seebeck coefficient for thermoelectric efficiency. The Generated Hypothesis uses Polyacrylamide (PAM) in place of Gelatin, which can provide a similar structural function but lacks some specific characteristics of Gelatin important for thermoelectric stability. It discusses ionic conductivity, paralleling that of KCl without directly employing the salt. Notably, it does not mention the redox pair, which is critical to the thermoelectric function as outlined in the Ground Truth Hypothesis.*

*However, while the Generated Hypothesis incorporates functional aspects related to ionic conductivity and chemical structure, it omits the crucial redox couple and does not fully replicate the essential components necessary for observable thermoelectric phenomena defined by the Ground Truth. The presence of PAM and evolving ionic conductivity offers a partial basis for plausible thermoelectric behavior under hypothetical conditions, but the complete omission of the redox pair is significant. The lack of adherence to all core key points and reliance on acceptable substitutions leads to a conclusion that while observable phenomena might still occur, they are limited. Given these evaluations, I assign $R = 0$ as there is a substantial gap in essential components for the reaction proposed in the Ground Truth Hypothesis.*

*Final Correction Factor 0 End*o

Our simulator analyzes Hypothesis 4, noting that enhancing the Seebeck coefficient of a polymer gel material requires a redox pair. The first three hypotheses incorporate redox pairs, either $[Fe(CN)_6]^{3-}$/$[Fe(CN)_6]^{4-}$ or $Fe^{3+}$/$Fe^{2+}$, which facilitate efficient conversion of thermal energy to electrical energy. In contrast, Hypothesis 4 only involves polymerizing acrylamide (PAM) and lacks a redox pair, rendering it unable to effectively convert thermal energy into electrical energy. Consequently, the thermoelectric potential (Seebeck coefficient, Se) is zero.

## C    EVALUATING THE SIMULATOR WITH REAL EXPERIMENT RESULTS

In this section, we present the validation of our simulator's accuracy using a dataset of 124 experimental hypotheses, detailing their classification and composition. We further compare the trends of the simulated results with the corresponding real experimental outcomes to assess the simulator's predictive performance and reliability in capturing real-world experimental behaviors.

### C.1    DATASET COMPOSITION AND ANALYSIS

To evaluate the performance of the simulator, we conducted a thorough analysis using real-world experimental data. We curated a set of 30 cutting-edge research questions, each designed to probe significant aspects of scientific research. These questions were carefully selected to encompass multiple areas within the scientific domain, ensuring a diverse and representative evaluation framework. Each question was associated with 3 to 6 hypotheses, resulting in a total of 124 authentic wet lab experiment results. This extensive dataset forms a robust foundation for assessing the simulator's predictive accuracy and reliability.

The 124 experiment results were sourced from key subfields of natural science to provide broad coverage of the discipline. The distribution of these results across subfields is presented in Table 5.

A statistical analysis of the 124 authentic wet lab results was conducted to rigorously evaluate the simulator's performance. By including a substantial number of experiments from various subfields, we ensured that the dataset captures a wide range of challenges encountered in experimental research. This approach minimizes potential biases from over-representing any single subfield, thereby strengthening the reliability of our evaluation. The dataset's diversity and scale provide a solid basis for assessing the simulator's ability to predict experimental outcomes accurately, offering valuable insights for future research and applications.

Table 5: Classification of the 124 real-world experiments used to validate the simulator.

| Category | Count |
|---|---|
| Energy Materials | 12 |
| Polymeric Materials | 8 |
| Applied Physics | 18 |
| Systems Biology | 10 |
| Organic Chemistry | 26 |
| Inorganic Chemistry | 24 |
| Analytical Chemistry | 26 |
| Total | 124 |

The use of authentic wet lab results bolsters the credibility of our findings. By grounding the evaluation in real experimental data, we ensured that the simulator's predictions were tested against the intricacies and variability of actual laboratory conditions. This approach not only validates the simulator's performance but also underscores its potential to guide subsequent research by delivering reliable and actionable predictions. The diverse dataset and representation of multiple subfields collectively contribute to a comprehensive and effective evaluation, paving the way for advancements in scientific simulation and experimentation.

## C.2 TREND COMPARISON WITH REAL EXPERIMENT RESULTS

To further assess the capabilities of our *CSX-Sim*, we utilized it to simulate 124 wet lab experiments. These experiments corresponded to 30 cutting-edge science questions, and their simulated outcomes were subsequently aggregated for a comprehensive analysis. For each of the 124 experiments, the simulated result was derived from the average of three trials conducted by *CSX-Sim*. These results, each corresponding to one of the curated questions, were systematically arranged in ascending order along the "Order of Experimental Results" axis, as depicted in Figure 6. This organization enabled a unified comparison between the simulated and actual experimental outcomes, with the vertical axis representing normalized experimental results to standardize the evaluation across the dataset.

Figure 6 compares the trends observed in *CSX-Sim* predictions (green line) with those from real experimental data (blue line). Error bars, representing the population standard deviation, illustrate the variability of the data points. Statistical significance was further established using the Bootstrap method, with results indicating ($p < 0.01$) (Berg-Kirkpatrick et al., 2012). The aggregated analysis reveals that the simulator effectively predicts the mean trends for all 30 sets of results, demonstrating a strong consistency with the mean of the actual experimental outcomes. This alignment of mean trends across the diverse questions underscores the simulator's ability to model scientific processes accurately, capturing the overall behavior of the experimental data, regardless of the specific subfield.

The use of normalized results ensures that differences in scale do not affect the comparison, allowing a fair assessment of the simulator's trend-matching capability. The close correspondence between the simulated and real mean data, as visualized in the figure, highlights the *CSX-Sim* broad applicability across the scientific domain. By successfully replicating the mean trends of the 124 results, the simulator proves to be a versatile tool, offering reliable predictions that can support a wide range of scientific research and applications.

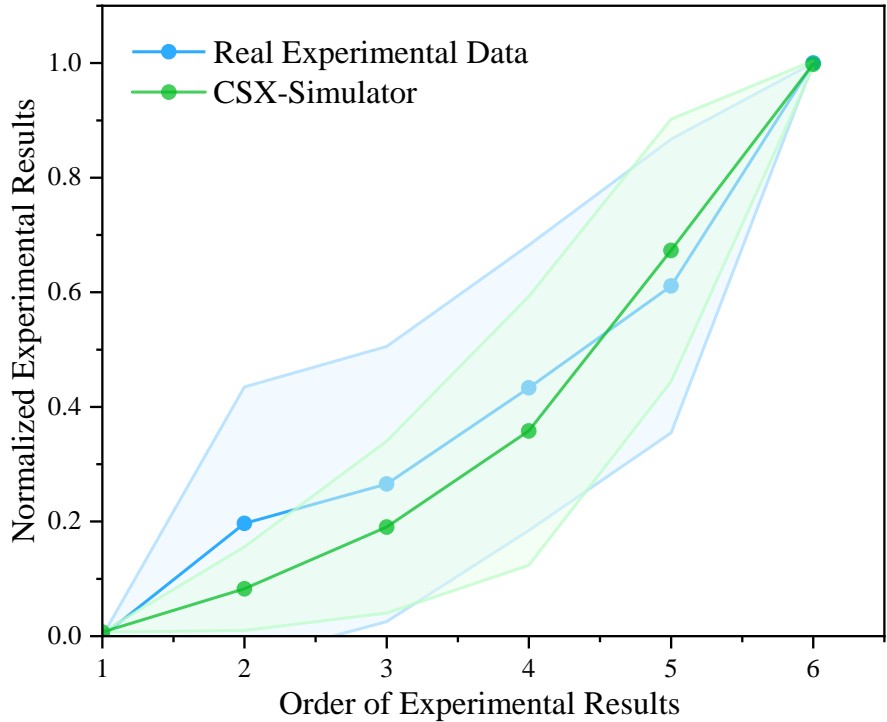

Figure 6: Comparison of simulated real experimental results with CSX-Simulator.

### C.3 Disciplines of the TOMATO-chem benchmark dataset

The TOMATO-chem benchmark dataset, spanning 12 distinct categories and totaling 3264 data records (Table 6), is a powerful resource due to its **inherent interdisciplinary nature** and wide scientific coverage[cite: 1627]. Far exceeding a focus on traditional chemical branches such as Organic, Inorganic, and Analytical Chemistry, the dataset features a strong emphasis on **advanced materials science** by including significant contributions from Energy Materials (363), Polymeric Materials (359), Metallic Materials (268), and Nanomaterials (316)[cite: 1640]. Furthermore, it effectively bridges fundamental research with practical applications through the inclusion of data from both **Chemical Engineering** (196) and **Environmental Engineering** (298)[cite: 1641]. Critically, the dataset extends its reach into the **chemical-biological interface**, incorporating samples from Molecular Biology (84) and Systems Biology (62)[cite: 1642]. This comprehensive disciplinary matrix underscores the dataset's utility as a robust benchmark for evaluating AI models across complex, intersecting scientific challenges, rather than isolated domain-specific problems.

## D Evaluation of Trend Alignment and Accuracy

### D.1 Evaluation of Trend Alignment

To quantitatively assess trend alignment between simulated and experimental results, we employed the *Spearman Rank Correlation Coefficient* (denoted as $\rho$). This non-parametric measure evaluates the monotonic relationship between the rankings of simulated and experimental outcomes, making it suitable for capturing trend consistency across diverse scientific problems.

The Spearman Correlation Coefficient is calculated as follows:

$$\rho = 1 - \frac{6 \sum d_i^2}{n(n^2 - 1)} \tag{8}$$

Where: $d_i$: The difference between the ranks of the $i$-th simulated and experimental result. $n$: The number of hypotheses in a given group (ranging from 3 to 6 per scientific question). $\rho$: The correlation

Table 6: Disciplinary classification of the TOMATO-chem benchmark dataset

| Category | Count |
|---|---|
| Energy Materials | 363 |
| Polymeric Materials | 359 |
| Metallic Materials | 268 |
| Nanomaterials | 316 |
| Applied Physics | 127 |
| Analytical Chemistry | 317 |
| Inorganic Chemistry | 392 |
| Organic Chemistry | 482 |
| Chemical Engineering | 196 |
| Environmental Engineering | 298 |
| Molecular Biology | 84 |
| Systems Biology | 62 |
| Total | 3264 |

coefficient, ranging from -1 (perfect negative correlation) to 1 (perfect positive correlation), with 0 indicating no monotonic relationship. A Spearman Correlation Coefficient ($\rho$) near 1 indicates strong trend alignment, meaning the simulated results closely mirror the relative ordering of experimental outcomes. Our *CSX-Sim* achieved a mean Spearman Correlation Coefficient of $\rho = 0.960$, significantly outperforming the baseline, as shown in Table 1, and demonstrating superior trend alignment.

To further assess the robustness of the simulator across diverse problems, we introduced the Perfect Consistency Indicator (PCI), a stringent metric that counts the number of question groups (out of the 30 scientific questions) where the simulated results achieved perfect trend alignment with the experimental results ($\rho = 1$). Perfect trend alignment requires an exact match in the ranking of simulated and experimental outcomes, making PCI a robust measure of the simulator's ability to consistently replicate experimental trends across all problems. Notably, our *CSX-Sim* achieved perfect trend alignment ($\rho = 1$) in 26 out of 30 question groups, significantly surpassing the baseline methods and highlighting its exceptional robustness and predictive fidelity.

### D.2 EVALUATION OF SIMULATOR ACCURACY

For evaluating prediction accuracy, we used the *Root Mean Square Error (RMSE)* to quantify the deviation between simulated and experimental values. The RMSE is defined as:

$$\text{RMSE} = \sqrt{\frac{1}{N} \sum_{i=1}^{N} (y_i - \hat{y}_i)^2} \tag{9}$$

Where: $y_i$: The experimental result for the $i$-th hypothesis. $\hat{y}_i$: The simulated result for the $i$-th hypothesis. The *CSX-Sim* exhibited a lower RMSE than the "Matched Score" baseline (Yang et al., 2025), signifying improved predictive accuracy, as substantiated by the results in Table 1.

To thoroughly evaluate the predictive accuracy of our simulator compared to real-world experimental outcomes, we tested its performance on a dataset of 124 authentic scientific hypotheses. For a comprehensive comparison, we calculated several performance indicators, as presented in Table 7. Building on the previously discussed metrics, we introduced three additional measures: Mean Squared Error (MSE), Mean Absolute Error (MAE), and Root Mean Squared Logarithmic Error (RMSLE). These metrics, defined below, enhance the robustness of our analysis by capturing different aspects of prediction error.

Table 7: Simulator validation against real-world wet-lab results.

| Simulator | MSE ($\downarrow$) | MAE ($\downarrow$) | RMSLE ($\downarrow$) |
|---|---|---|---|
| Matched Score | 0.068 | 0.179 | 0.166 |
| *CSX-Sim* | **0.058** | **0.161** | **0.147** |
| w/o CriticalPoints | 0.064 | 0.174 | 0.159 |
| w/o ComponentExtraction | 0.087 | 0.215 | 0.192 |

Below, we define each metric used in the evaluation, along with their respective formulas, to ensure scientific rigor:

Mean Squared Error (MSE): MSE measures the average squared difference between predicted values $\hat{y}_i$ and actual values $y_i$ across $n$ samples. It is defined as:

$$\text{MSE} = \frac{1}{n} \sum_{i=1}^{n} (\hat{y}_i - y_i)^2 \tag{10}$$

A lower MSE indicates higher predictive accuracy, with larger errors penalized more heavily due to squaring.

Mean Absolute Error (MAE): MAE quantifies the average absolute difference between predicted and actual values, calculated as:

$$\text{MAE} = \frac{1}{n} \sum_{i=1}^{n} |\hat{y}_i - y_i| \tag{11}$$

This metric is less sensitive to outliers than MSE, providing a more balanced measure of error.

Root Mean Squared Logarithmic Error (RMSLE): RMSLE focuses on relative errors by evaluating the logarithmic difference between predicted and actual values:

$$\text{RMSLE} = \sqrt{\frac{1}{n} \sum_{i=1}^{n} \left(\log(\hat{y}_i + 1) - \log(y_i + 1)\right)^2} \tag{12}$$

This metric is particularly useful for datasets with exponential trends or varying error scales.

As shown in Table 7, *CSX-Sim* consistently outperforms the "Matched Score" baseline (Yang et al., 2025) across all metrics, achieving an MSE of 0.058, an MAE of 0.161, and an RMSLE of 0.147. Ablation studies further reveal the contributions of individual components: the removal of Critical-Points results in a slight performance decline (MSE of 0.064, MAE of 0.174, RMSLE of 0.159), while the exclusion of ComponentExtraction leads to more significant degradation (MSE of 0.087, MAE of 0.215, RMSLE of 0.192). These results underscore the importance of both critical point identification and component extraction in achieving high predictive accuracy and robustness in simulation outcomes.

## E  DIFFERENT LEVELS OF DISTORTION

We collaborated with Scientific PhD students to identify and design three common types of distortions encountered in scientific research: local maxima/minima, plateaus, and cliffs. These distortion patterns reflect typical challenges in hypothesis evaluation, drawing on domain expertise and established heuristics to ensure relevance. We defined three distinct distortion levels—Simple Noise, Moderate Noise, and Complex Noise—and incorporated them into the hypothesis embedding function $\phi(\cdot)$ to simulate increasingly challenging feedback conditions.

In scientific scientific hypotheses, biases in understanding key factors can result in specific distortion patterns. For instance, when adding guanidine sulfate to polymer thermoelectric materials, recognizing it solely as a salt providing hydrogen bonds for the reaction—while overlooking its influence on the entropy of redox pairs—can lead to a local maximum, as this oversight may enhance thermoelectric performance unexpectedly. Similarly, misjudging irrelevant factors, such as additives

in organic reactions with no actual impact, can create a plateau effect. Conversely, misjudging critical factors, like the temperature's role in enzyme activity during enzyme studies, can produce a cliff if the temperature is incorrectly assumed to inhibit the reaction entirely. These elements—local maxima/minima, plateaus, and cliffs—present significant challenges in optimization problems within scientific research.

Through extensive discussions with scientific experts, we conducted a statistical analysis to evaluate the discrepancies between wet lab results and empirical expected outcomes across diverse experimental scenarios. This process enabled us to statistically analyze the frequency of the three types of distortions—local maxima/minima, plateaus, and cliffs—across various scientific scenarios. We then quantified the occurrence of these distortions in different scenarios and sorted them by frequency, from low to high. Based on this distribution, we categorized the discrepancies: the top 35% of observed gaps were classified as Simple Noise, the middle 40% as Moderate Noise, and the bottom 25% as Complex Noise. Furthermore, we integrated the three distortion levels—Simple Noise, Moderate Noise, and Complex Noise—into the hypothesis embedding function $\phi(\cdot)$ to simulate increasingly challenging feedback conditions. This structured stratification provided a clear framework to evaluate the varying impacts of different scenarios on our simulator, facilitating a deeper understanding of the simulator's performance under diverse conditions.

Table 8: The composition of different types of noise.

| Noise Conditions | Local Maxima/Minima | Plateaus | Cliffs |
|---|---|---|---|
| Simple | 0-10 | 0-2 | 0-2 |
| Medium | 0-30 | 0-6 | 0-6 |
| Complex | $\geq 30$ | $\geq 3$ | $\geq 3$ |

These distortions, along with their detailed quantities, are outlined in the accompanying Table 8, which illustrates the composition of different types of noise across various conditions. For instance, simple noise conditions are associated with 0-10 local maxima/minima, 0-2 plateaus, and 0-2 cliffs. Medium noise conditions escalate these figures to 0-30 local maxima/minima, 0-6 plateaus, and 0-6 cliffs. In complex noise scenarios, the challenges intensify, with $\geq 30$ local maxima/minima, $\geq 3$ plateaus, and $\geq 3$ cliffs, reflecting the increased difficulty in achieving optimal solutions. We constructed three distinct noise levels to evaluate the robustness of our *CSX-Rank* under complex scientific feedback conditions.

By comparing Table 3, we observed that with the introduction of noise, the experiment-guided ranking method requires a significantly higher number of simulation feedback iterations to identify the ground truth scientific hypothesis as the complexity of the noise increases. This is primarily due to the growing discrepancy between highly complex noise and real experimental feedback, where simulation feedback contains substantial erroneous information, thereby degrading the performance of screening the ground truth scientific hypothesis from the generated scientific hypotheses.

# F    PERFORMANCE COMPARISON OF DIFFERENT FUNCTIONS IN THE SIMULATOR

Table 9: Performance comparison of different functions in the simulator.

| **Function** | **Spearman Corr. ($\uparrow$)** | **RMSE ($\downarrow$)** | **Perfect Consistency ($\uparrow$)** |
|---|---|---|---|
| Linear Function | 0.9708 | 0.1959 | 24/30 |
| Gaussian Function | 0.9600 | 0.2147 | 26/30 |
| Absolute Value Function | 0.9626 | 0.2595 | 23/30 |
| Quadratic Function | 0.9682 | 0.3996 | 22/30 |

This supplementary study was conducted to validate the robustness of our core mathematical modeling. While the Gaussian function was selected for its well-behaved mathematical properties and intuitive alignment with our core assumptions, the framework's success is not tied to any single function form. The choice of function can be viewed as a tunable hyperparameter.

Table 9 presents the results of a comparative study of various monotonic functions serving as the core of the simulator. Performance was assessed on Spearman Correlation ($\uparrow$), RMSE ($\downarrow$), and Perfect Consistency ($\uparrow$). The results show that all tested functions provide effective ranking guidance, which underscores the framework's overall robustness. This analysis confirms that our framework is adaptable and can accommodate different function forms, enhancing its generalizability across domains.

## G  RELATED WORK

Most prior work on hypothesis ranking has focused on pre-experiment ranking. Some approaches assign a score to each hypothesis and rank them accordingly, providing a simple and efficient solution (Yang et al., 2024; 2025; Zhou et al., 2024). Others adopt a pairwise ranking strategy, evaluating hypothesis pairs one at a time (Si et al., 2024; Liu et al., 2025). However, these methods rely solely on the internal reasoning of LLMs and do not incorporate feedback from experimental outcomes. To our knowledge, few existing works leverage experimental feedback in hypothesis-driven tasks, and those that do are confined to domains with highly efficient verifiers, enabling rapid hypothesis testing and direct refinement rather than explicit ranking. Notably, recent methods in mathematics (Romera-Paredes et al., 2024; Shojaee et al., 2024; Ma et al., 2024) and programming (Novikov et al., 2025; Qiu et al., 2024) incorporate feedback loops by refining hypotheses based on verification outcomes. In contrast, our work targets natural science domains, where real experiments are far more costly, rendering such exhaustive trial-and-error strategies impractical. This motivates the need for a more deliberate experiment-guided ranking process, designed to maximize the information gained from each costly experiment when prioritizing future hypotheses. Roohani et al. (2024) explore hypothesis generation in a genetic perturbation setting, where task-specific feedback can be computed directly (e.g., via gene overlap). This remains a niche domain where efficient verifiers are available. By contrast, our work focuses on constructing general-purpose simulators, enabling the study of experiment-guided ranking in settings where real experiments are costly and feedback is scarce.

## H  EVALUATION OF EXPERIMENT-GUIDED RANKING AND ITS SOCIETAL BENEFITS

The intricate web of scientific knowledge, combined with the multitude of factors influencing hypothesis analysis, often leads to the gradual accumulation of small cognitive biases. These biases can significantly distort the final experimental outcomes, creating substantial disparities between expected and observed results. To address this challenge, we conducted a comparative analysis between two distinct approaches: the experiment-guided ranking method, which leverages simulation feedback or real experimental results to refine hypothesis selection, and the pre-experiment method, which relies solely on the model's prior knowledge for screening the ground truth hypothesis. Our findings reveal that the experiment-guided ranking method demonstrates a marked improvement over its counterpart. By integrating simulation feedback, this method allows for a reflective process that considers previous simulation (and experimental) results. This iterative reflection provides more contextually relevant information, enabling the selection of the next hypothesis with greater precision. Consequently, this approach effectively mitigates the accumulation of biases, thereby enhancing the efficiency and accuracy of experimental screening processes.

The ranking of hypotheses emerges as a pivotal element in automated scientific discovery, particularly in natural sciences, where wet-lab experiments are costly and are constrained by low throughput. Traditional approaches, such as pre-experiment ranking, depend exclusively on the internal reasoning of large language models, lacking integration with empirical experimental outcomes. In contrast, we introduce the novel task of experiment-guided ranking, designed to prioritize candidate hypotheses by leveraging insights from previously tested results. However, the development of such strategies is hindered by the impracticality of repeatedly conducting real experiments in natural science domains due to time, cost, and resource limitations. To overcome this obstacle, we propose a simulator grounded in three domain-informed assumptions, modeling hypothesis performance as a function of its similarity to a known ground truth hypothesis, with performance perturbed by noise to reflect real-world variability. To validate this simulator, we curated a dataset comprising 124 scientific hypotheses, each accompanied by experimentally reported outcomes, providing a robust foundation for evaluation.

Building on this simulator, we developed a pseudo experiment-guided ranking method that clusters hypotheses based on shared functional characteristics and prioritizes candidates using insights derived from simulated experimental feedback. Our experimental results demonstrate that this method outperforms both pre-experiment baselines and strong ablations, highlighting its potential to revolutionize hypothesis selection in scientific research. Beyond academic and scientific advancements, this approach holds promising societal impacts. By reducing the need for extensive wet-lab experiments, it can lower research costs and accelerate the development of new materials and drugs, potentially improving healthcare access and environmental sustainability. Additionally, the enhanced efficiency in hypothesis testing could foster innovation in industrial applications, such as cleaner energy solutions, contributing to global efforts to address climate change and promote sustainable development.

## I    SCALABILITY ANALYSIS

Table 10: Number of experiments required to identify the ground truth hypothesis across methods.

| Method | Trials (N = 64) | Trials (N = 128) |
|---|---|---|
| Uninformed Search | 32.5 | 64.5 |
| Pre-experiment ranking | 28.6 | 51.3 |
| **CSX-Rank (ours)** | **15.2** | **30.7** |

To evaluate the scalability of our proposed method, we expanded the pool of candidate hypotheses from N=64 to N=128. The results, presented in Table 10, show that our method, *CSX-Sim*, required 15.2 trials for 64 candidates and 30.7 trials for 128 candidates. In contrast, Uninformed Search and Pre-experiment ranking required 64.5 and 51.3 trials, respectively, for the larger candidate pool. The experimental results align with our theoretical analysis. The performance of *CSX-Rank* demonstrates a near-linear growth, which is consistent with its average-case time complexity of $O(N)$. The observed slope of approximately $0.24 \times N$ confirms this scalability and shows that *CSX-Rank* retains a substantial cost advantage even as the candidate pool expands. Theoretically, in a best-case scenario where the clustering of hypotheses is highly effective, the cost could be reduced to $O(\log N)$, as evidence from one experiment can be generalized to every hypothesis within its cluster.

## J    DISCUSSION ON THE RELATIONSHIP WITH ACTIVE LEARNING AND SIM2REAL

### J.1    OPERATIONAL PARADIGM: TRAINING-FREE REASONING VS. MODEL RETRAINING

Standard Active Learning strategies, widely applied in domains such as nanocatalysis Perumal et al. (2025) and drug discovery Borkowski et al. (2020); van Tilborg & Grisoni (2024), typically rely on a *retraining-based* workflow. In these settings, the model must be periodically updated or fine-tuned using newly acquired experimental labels to improve its predictive boundaries. Similarly, traditional Sim2Real methods often necessitate the collection of real-world data to fine-tune policies and bridge the domain gap via gradient updates Wagenmaker et al. (2024).

In contrast, our approach operates under a **Training-Free Paradigm** powered by In-Context Reinforcement Learning (ICRL). By keeping the underlying Large Language Model (LLM) frozen, our agent performs reasoning and hypothesis prioritization from the very first trial without the computational overhead or data requirements of gradient-based updates. This is particularly critical in wet-lab settings where data is extremely scarce (often $N < 100$) and high-fidelity fine-tuning is impractical.

### J.2    OBJECTIVE: OPTIMIZATION-CENTRIC VS. GENERALIZATION-CENTRIC

The primary objective of classical AL in physics Ding et al. (2023) and chemical space exploration Smith et al. (2018); Khalak et al. (2022) is often rooted in *Generalization*. These methods typically employ acquisition functions designed to reduce global uncertainty across the entire search space, aiming to learn a model that performs well on the underlying distribution.

Conversely, our framework is strictly **Optimization-Centric**. As defined in our objective function (Equation 6), we are not concerned with reducing uncertainty in irrelevant regions of the chemical space. Instead, our agent focuses solely on identifying the optimal hypothesis $h^*$ with the minimum number of trials. This distinct focus allows our method to be more aggressive in exploitation, prioritizing candidates that are likely to yield high performance rather than those that merely improve model robustness.

### J.3 INTERPRETABILITY: STRUCTURE-AWARE REASONING VS. SCALAR ACQUISITION

A significant limitation of traditional AL is the opacity of its decision-making process. Candidates are typically selected based on scalar acquisition functions (e.g., Upper Confidence Bound (UCB) or Expected Improvement) Settles (2009); Smith et al. (2018), which offer little insight into *why* a specific candidate is promising.

Our framework addresses this by employing **Structure-Aware Reasoning**. As detailed in Section §3.2, our agent prioritizes candidates through explicit component clustering and reasoning steps. This allows domain experts to audit the decision logic, understanding not just the numerical rank of a hypothesis, but the chemical rationale behind its selection. This "White-Box" approach fosters greater trust and collaboration between AI agents and human scientists.

Table 11: Comparison between Standard Active Learning (AL) / Sim2Real and Our Proposed Framework.

| Feature | Standard AL / Sim2Real | Ours (Experiment-Guided Ranking) |
|---|---|---|
| **Paradigm** | **Training-Dependent:** Requires iterative re-training or fine-tuning (Gradient-based). | **Training-Free:** Uses In-Context RL with a frozen model (Gradient-free). |
| **Data Req.** | High data demand to update model weights effectively. | Effective in extreme data scarcity (Few-shot/Zero-shot). |
| **Objective** | **Generalization:** Reduces global uncertainty; learns the landscape Ding et al. (2023); Khalak et al. (2022). | **Optimization:** Directly targets the optimal hypothesis $h^*$ (Equation 6). |
| **Interpretability** | **Opaque:** Based on scalar values (e.g., UCB scores) Settles (2009). | **Transparent:** Based on component reasoning and clustering. |

## K SENSITIVITY ANALYSIS ON MODEL ARCHITECTURES AND ENVIRONMENTAL NOISE

To demonstrate the robustness of our framework, we conducted a comprehensive sensitivity analysis focusing on two critical dimensions: the choice of Large Language Model (LLM) backbones and the impact of simulator noise levels.

### K.1 ROBUSTNESS ACROSS LLM ARCHITECTURES

We evaluated the generalizability of our component-based reasoning framework by varying the underlying models for both the simulator environment and the agent's policy. This ablation ensures that our performance gains are not derived from overfitting to a specific model family.

Table 12 presents the average number of trials required to identify the optimal hypothesis under different configurations. The results indicate that:

- **Model Agnosticism:** Our method consistently outperforms the Baseline (Pre-Experiment Ranking) regardless of the model combination used (e.g., using Gemini 2.5 Flash-Lite as the simulator).

- **Scaling with Capability:** Replacing the policy model with a stronger reasoning engine (Claude 3 Sonnet) further reduces the average trials to 14.12. Notably, Claude 3 Sonnet has a knowledge cutoff of August 2023, ensuring zero data contamination against our 2024-2025 validation dataset.

These findings confirm that the core advantage of our approach lies in the structural reasoning framework rather than the parametric knowledge of a specific LLM.

Table 12: Performance Comparison Across Different LLM Backbones. Our framework exhibits consistent robustness and scales effectively with stronger reasoning models.

| Configuration (Simulator / Policy) | Avg. Trials | Improvement |
|---|---|---|
| Baseline (Pre-Experiment Ranking) | 28.60 | - |
| Ours (Gemini 2.5 Flash-Lite / GPT-4o-mini) | 19.56 | **+31.6%** |
| Ours (GPT-4o-mini / GPT-4o-mini) | 15.20 | **+46.8%** |
| Ours (GPT-4o-mini / Claude 3 Sonnet[*]) | **14.12** | **+50.6%** |

[*]Claude 3 Sonnet knowledge cutoff: Aug 2023 (ensuring zero contamination).

### K.2 Resilience to Simulator Noise Assumptions

Beyond model architecture, we stress-tested the policy's resilience to environmental stochasticity. As detailed in the main text (Section 4.4), we introduced varying levels of signal distortion—Simple, Medium, and Complex—to simulate experimental error and simulator inaccuracies.

While the absolute efficiency naturally correlates with signal fidelity, our method (CSX-Rank) demonstrates effective mitigation of misleading signals. Even under the "Complex" noise regime, our approach significantly outperforms ablated variants (e.g., achieving 32.7 trials vs. 40.5 trials for the baseline), proving that the structural analysis module effectively filters noise and prevents the agent from being misled by individual erroneous data points.

## L    Analysis of Failure Modes and Handling of Emergent Properties

In this section, we provide a critical analysis of the specific scenarios where our framework may diverge from ground truth, followed by a theoretical discussion on how the component-based approach addresses emergent scientific phenomena.

### L.1    Case Study on Failure Modes: Complex Physicochemical Mechanisms

While CSX-Rank demonstrates high accuracy in general retrieval, failures can occur when the underlying scientific mechanism relies on subtle, high-order physical interactions rather than direct compositional effects.

A concrete instance of divergence was observed in the domain of **Thermoelectric Materials** . The ground truth hypothesis introduced $(\mathbf{Gdm})_2\mathbf{SO}_4$ to interact with the $\mathbf{K}_3\mathbf{Fe(CN)}_6/\mathbf{K}_4\mathbf{Fe(CN)}_6$ redox pair. Theoretically, this addition boosts the Seebeck coefficient ($S_e$) by significantly increasing the reaction entropy difference ($\Delta S_{rc}$), governed by the thermodynamic relation:

$$S_e = \frac{\Delta E}{\Delta T} = \frac{\Delta S_{rc}}{nF} \tag{13}$$

Furthermore, the system involves complex transport dynamics governed by the **Eastman entropy of transfer** ($\hat{S}_i$), which drives the thermal diffusion potential ($S_{td}$):

$$S_{td} = \frac{\sum_i q_i n_i^0 \hat{S}_i D_i}{\sum_i q_i^2 n_i^0 D_i} \tag{14}$$

**Failure Analysis:** In this specific case, the ranking policy correctly identified the key component but misinterpreted its functional role. The model overlooked these intricate entropic contributions ($\Delta S_{rc}$ and $\hat{S}_i$) and instead classified the sulfate solely as a salting-out agent for **mechanical reinforcement**. This misinterpretation led to an underestimation of the hypothesis's rank. This suggests that while the framework excels at structural and functional matching, it may struggle with mechanisms involving implicit higher-order thermodynamic derivatives.

## L.2   HANDLING EMERGENT AND NON-DECOMPOSABLE PROPERTIES

A common challenge in scientific AI is the assumption that properties are the sum of their parts. We address non-compositional and emergent properties through a combination of mathematical gating and functional abstraction.

### L.2.1   MATHEMATICAL HANDLING VIA MULTIPLICATIVE GATING

Our scoring model (Equation 3) is designed to move beyond simple additive similarity. Crucially, it incorporates a multiplicative gating term:

$$\text{Score}(h) \propto \prod_{i \in \mathcal{C}} \mathbf{1}(s_i > 0) \tag{15}$$

This term ensures that if a "Critical Point" (a necessary condition for the phenomenon to emerge) is missing, the total score drops to zero. This effectively models "all-or-nothing" emergent behaviors often seen in physics and biology, preventing the system from highly ranking a hypothesis that has many good peripheral features but lacks the core mechanism.

### L.2.2   FUNCTIONAL ABSTRACTION VS. ATOMIC DECOMPOSITION

Our decomposition strategy (Figure 3) operates at the **functional mechanism level** rather than the atomic level. This mirrors how human scientists conceptualize emergent phenomena. For example, in high-temperature superconductors, a hypothesis is not decomposed into raw atoms (Cu, O, La), but into functional units:

- **Cuprate Planes:** Responsible for the superconducting mechanism.
- **Charge Reservoirs:** Responsible for doping levels.
- **Interlayer Spacing:** Responsible for strain and critical temperature ($T_c$) modulation.

By abstracting these emergent behaviors into distinct functional units, the "emergence" is effectively encapsulated within the component, allowing the linear ranking framework to remain valid. However, we acknowledge limitations in purely abstract theoretical physics where such functional decomposition is less applicable.

### L.2.3   ROBUSTNESS TO NON-LINEAR LANDSCAPES

To empirically validate this, we introduced "Landscape Cliffs" in our distortion experiments (Section 4.4). These cliffs simulate scenarios where a small change in similarity results in a drastic drop in performance (a hallmark of non-linear systems). Our results demonstrate that CSX-Rank maintains high ranking efficiency even in these non-smooth regimes, confirming that the combination of functional abstraction and critical gating provides robustness against non-linear complexities.

## M   THEORETICAL ANALYSIS OF SEARCH COMPLEXITY REDUCTION VIA FUNCTIONAL DECOMPOSITION

In this section, we provide a formal analysis of how the *Experiment-Guided Ranking* framework, driven by *Functional Decomposition*, fundamentally reduces the search complexity compared to traditional Pre-experiment Ranking baselines. We model the hypothesis discovery process as an optimization problem.

### M.1 PROBLEM FORMULATION

**Definition 1 (Hypothesis Space and Component Space).** Let $\mathcal{K} = \{k_1, k_2, \ldots, k_m\}$ be the universal discrete set of functional component clusters, where $K = |\mathcal{K}|$ is the total number of unique functional modules available in the domain. We define a Hypothesis $h$ as a composition of functional modules selected from $\mathcal{K}$. The Hypothesis Space $\mathcal{H}$ is the set of all valid combinations bounded by a maximum structural complexity $m$:

$$\mathcal{H} = \{h = \{k_1, \ldots, k_n\} \subseteq \mathcal{K} \mid n \leq m\} \tag{16}$$

Assuming a standard combinatorial setting, the cardinality of the hypothesis space scales exponentially with the complexity $m$:

$$N = |\mathcal{H}| \approx \sum_{n=1}^{m} \binom{K}{n} \approx \mathcal{O}(K^m) \tag{17}$$

where $N$ represents the total number of candidate hypotheses.

## M.2 BASELINE COMPLEXITY: BLACK-BOX SEARCH

In traditional Pre-experiment Ranking or Naive Search, hypotheses are treated as Atomic Black Boxes. The internal structure is opaque to the selection policy $\pi$, relying on holistic evaluation.

**Proposition 1 (Information Isolation).** Without decomposition, the feedback from an experiment on hypothesis $h_i$, denoted as $y_i = f(h_i)$, provides information strictly limited to the joint probability $P(h_i)$. The search policy approximates Random Sampling without Replacement in the high-dimensional space. To identify the optimal hypothesis $h^*$, the expected number of trials $\mathbb{E}[T_{atomic}]$ scales linearly with the size of the hypothesis space:

$$\mathbb{E}[T_{atomic}] \propto N \approx \mathcal{O}(K^m) \tag{18}$$

**Conclusion:** The baseline suffers from the Curse of Dimensionality, yielding Exponential Complexity with respect to the hypothesis structural complexity $m$.

## M.3 CSX-RANK COMPLEXITY: MODULAR OPTIMIZATION

Our approach leverages Functional Decomposition, transforming the problem from identifying the optimal combination $h^*$ to identifying the set of optimal functional modules $\mathcal{K}^* \subset \mathcal{K}$.

**Proposition 2 (Feedback Propagation and Pruning).** Crucially, CSX-Rank attributes the experimental outcome $y$ to the **marginal utility** of individual modules $k \in \mathcal{K}$. A negative feedback on a module $k_{bad}$ allows the agent to update the module-level posterior $P(k_{bad}|y)$ and effectively prune not just the tested hypothesis, but the entire subset of hypotheses $\mathcal{H}_{sub} \subset \mathcal{H}$ containing $k_{bad}$:

$$\mathcal{H}_{sub} = \{h \in \mathcal{H} \mid k_{bad} \in h\} \tag{19}$$

**Complexity Analysis.** The efficiency of this reduction relies on the degree to which the target property can be decomposed. We analyze the asymptotic complexity across three regimes:

- **Best-case ($\mathcal{O}(K)$):** In ideal scenarios where hypotheses are perfectly functionally modular (i.e., the validity of a hypothesis is the sum/conjunction of independent valid components), the agent essentially performs a parallel screening of the component space $\mathcal{K}$. The complexity scales linearly with the number of unique modules $K$, as $K \ll N$.

- **Average-case:** In practical settings, while component interactions introduce noise, the search complexity is fundamentally governed by the effective number of functional modules ($m$) per hypothesis, not the raw count of candidates ($N$). While baselines must exhaustively search the combinatorial space ($N \approx K^m$), our method's cost grows proportionally to the diversity of mechanisms needed to cover the space. The observed slope in Table 10 ($0.24 \times N$) reflects this advantage: even as $N$ doubles (implying a vast increase in combinations), our trials grow slowly, bounded by the component-wise learning rate rather than combinatorial enumeration.

- **Worst-case ($\mathcal{O}(K^m)$):** In scenarios with strong entanglement or pure emergence, feedback on individual components provides zero information gain ($P(k|y) \approx P(k)$). In this limit, the strategy degenerates to the baseline Black-box Search, recovering the exponential combinatorial complexity.

**Applicability and Limitations.** This theoretical boundary highlights that our decomposition-based framework is particularly well-suited for Experimental Sciences (e.g., Chemistry, Biology, Materials Science). In these domains, the scientific principle "Structure determines Property" often implies that specific functional motifs (e.g., a benzene ring, a specific protein domain) carry intrinsic, transferable utility. Conversely, we acknowledge limitations in domains like Theoretical Physics or highly abstract

mathematical derivation tasks. In such fields, properties are often strictly emergent or governed by high-order non-linear equations where decomposing a hypothesis into "functional atoms" destroys the semantic meaning, making the Best-case complexity unattainable.

