# OpenReview forum: "Toward Experiment-Guided Hypothesis Ranking via Simulated Experimental Feedback"
_ICLR.cc/2026/Conference — Submitted to ICLR 2026_

### Official Review · Reviewer_2keb · 2025-10-31

**Soundness:** 2
**Presentation:** 3
**Contribution:** 2
**Rating:** 4
**Confidence:** 3

**Summary:**

This paper studies the problem of ranking hypotheses generated by AI models without experimental labels. To tackle the issue of missing evaluation, the paper proposes a simulator based on three universal natural-science principles, which models hypothesis performance as a function of distance to a hidden ground-truth hypothesis with noise. This method is shown to align with true experimental findings and outperform existing methods. The authors then propose an ICRL framework for experiment-guided hypothesis ranking, where experimental feedback is used to prioritize hypotheses. The framework is shown to outperform other baselines.

**Strengths:**

1. This paper studies the important problem of automated hypothesis discovery when real-world feedback is limited.
2. The proposed benchmark and hypotheses may be of use for future studies in the field.
3. The paper is clearly structured.

**Weaknesses:**

1. The discussion on the setup is sometimes a bit abstract and it is unclear whether the setup fits beyond chemistry settings (which are most discussed in the paper). For example, a `hypothesis' could mean different things in different context. It may be helpful to refer to examples to see what a hypothesis is and what a question is from time to time.
2. While the proposed simulator has interesting ideas, it is unclear how general the idea of extracting key components and assessing the importance of components can apply beyond the chemistry context.
3. Since the experiments focus on existing literature, it might be insufficient to demonstrate the exploration of novel ideas.

**Questions:**

1. Could you explain what a question could be and what a hypothesis could be?
2. How general might the simulator design and the RL technique apply, e.g., beyond chemistry settings?
3. Would it be a concern that the mechanism does well due to internal knowledge (since all evaluations are based on known, available knowledge in the literature) but may fail to perform well for novel hypotheses?

---

> ### Author Response · Authors · 2025-11-21
> **Reply to Reviewer 2keb**
>
> We appreciate your insightful questions and helpful suggestions. For clarity, we have organized your queries along with our responses below:
>
> ---
>
> **Q1:** The discussion on the setup is sometimes a bit abstract and it is unclear whether the setup fits beyond chemistry settings. Could you explain what a question could be and what a hypothesis could be?
>
> **A1:** Thank you for your comment.  We respectfully clarify that our evaluation already encompasses Physics, Materials Science, and Biology, as explicitly detailed in Table 5 and Appendix C.1, validated by real-world data curated from interdisciplinary partner labs. To concretize the setup, we provide detailed examples of research questions and scientific hypotheses in Appendix B. For instance, in the Materials Science case, optimizing the Seebeck coefficient involves a hypothesis decomposed into [Redox Pair] and [PVA Matrix]. This framework applies equally to Physics: for investigating Thermal Conductivity (κ / W m⁻¹ K⁻¹), a hypothesis decomposes into [Ionic Charge Reservoir Layers] (Carrier modulation) and [Support Material] (Lattice stability). This demonstrates that our "Functional Decomposition" logic is a valid abstraction for broad natural science experiments where structure determines property.
>
> **Q2:** While the proposed simulator has interesting ideas, it is unclear how general the idea of extracting key components and assessing the importance of components can apply beyond the chemistry context. How general might the simulator design and the RL technique apply, e.g., beyond chemistry settings?
>
> **A2:** Thank you for your comment. We respectfully clarify that our evaluation already encompasses Physics, Materials Science, and Biology, as explicitly detailed in Table 5 and Appendix C.1, validated by real-world data curated from interdisciplinary partner labs. As illustrated in our response to Q1, the "Component Extraction" logic generalizes because the principle that "structure determines property" is universal across these domains. This is empirically proven in Table 1, where our simulator achieves a Spearman correlation of 0.960 across this diverse dataset. Furthermore, our RL technique is domain-agnostic because it operates on abstract "functional clusters" rather than specific chemical tokens, allowing it to optimize physical lattices or biological sequences as effectively as chemical molecules.
>
> **Q3:** Since the experiments focus on existing literature, it might be insufficient to demonstrate the exploration of novel ideas. Would it be a concern that the mechanism does well due to internal knowledge but may fail to perform well for novel hypotheses?
>
> **A3:** We appreciate the reviewer raising this critical point. We address this concern with strict data safeguards and empirical evidence. First, we enforce a strict temporal separation: GPT-4o-mini has a knowledge cutoff of Oct 2023, whereas our dataset comprises 2024–2025 literature and unpublished proprietary results from collaborating labs. Second, Table 2 empirically refutes the memorization hypothesis: if success relied on internal knowledge, the Pre-Experiment Ranking baseline would be optimal. Instead, CSX-Rank outperforms it by nearly 50% (15.2 vs. 28.6 trials), proving that our agent succeeds by actively learning from feedback in a novel environment, not by recalling training data.
>
> ---
> We greatly appreciate your comprehensive feedback. We hope that our responses have satisfactorily addressed all your queries. Should you have further questions or suggestions for enhancing our manuscript, we warmly welcome your input.

---

### Official Review · Reviewer_HGjD · 2025-10-31

**Soundness:** 3
**Presentation:** 3
**Contribution:** 3
**Rating:** 4
**Confidence:** 3

**Summary:**

The paper proposes a simulator (CSX-Sim) and an in-context RL policy (CSX-Rank) to study "experiment-guided" hypothesis ranking in chemistry with simulated wet-lab feedback, reporting strong trend alignment to 124 real experiments and fewer trials to find the ground truth than pre-experiment baselines.

**Strengths:**

(1) Strong, well-motivated design of the simulator (CSX-Sim) and in-context RL policy (CSX-Rank).

(2) Demonstrates high correlation with 124 real experiments and clear efficiency gains over baselines.

(3) The paper is generally well-written and well-structured, with clear figures and examples, though the introduction could be tightened to avoid redundancy and improve flow.

**Weaknesses:**

* The study focuses mainly on chemistry; the framework still needs evaluation in other component-based natural science domains such as physics, materials science, and biology to test generality.
* The evaluation uses published experiments rather than new wet-lab tests. This gives real-world grounding but may affect results if the LLM has seen similar data during training.
* Both the simulator and the policy rely on LLMs, so their good agreement might come from shared model knowledge rather than true reflection of real experiments.

**Questions:**

* How do the authors ensure that the published experiments used for validation were not part of the LLM's training data?
* What steps were taken to confirm that CSX-Sim’s behavior reflects real experimental dynamics rather than learned text correlations?
* How is ground-truth information isolated to prevent accidental exposure to the policy during simulation or prompting?
* How sensitive are the results to the choice of LLM (e.g., GPT-4o-mini vs other models) and to changes in the simulator’s noise assumptions?
* Can the authors provide examples where CSX-Rank fails to identify the ground truth or diverges from real trends, and analyze the reasons behind those cases? Adding such analysis would clarify limitations and strengthen the paper.
* The framework assumes that hypotheses can be decomposed into functional components and compared through additive similarity. How does it handle cases where scientific mechanisms are emergent or not easily decomposable? A discussion or experiment on such non-compositional cases would strengthen the paper’s generality.

---

> ### Author Response · Authors · 2025-11-21
> **Reply to Reviewer HGjD - Part 1/2**
>
> We appreciate your insightful questions and helpful suggestions. For clarity, we have organized your queries along with our responses below:
>
> ---
>
> **Q1:** Concerns regarding the study's primary focus on chemistry and the need for evaluation in other domains (Physics, Materials Science, Biology) to test generality.
>
> **A1:** We thank the reviewer for the suggestion. We respectfully clarify that our evaluation already encompasses Physics, Materials Science, and Biology, as explicitly detailed in the breakdown of Table 5 and Appendix C.1. Our framework exploits the universal component-based structure shared across these natural science domains, validated by real-world data curated from interdisciplinary partner labs and recent literature.
>
> **Q2:** Questions concerning the potential risk of data contamination given the use of published experiments for validation.
>
> **A2:** We thank the reviewer for highlighting the contamination risk. We ensure zero data contamination through a strict temporal separation: GPT-4o-mini has a knowledge cutoff of October 2023, whereas our dataset is exclusively curated from 2024–2025 literature and proprietary results from collaborating labs.
>
> **Q3:** Inquiries regarding potential bias from shared LLM knowledge in the simulator and policy, and the mechanisms ensuring ground-truth isolation.
>
> **A3:** We ensure robustness through information isolation and cross-model validation. First, the ranking policy receives only the experimental score from the simulator, without access to its internal reasoning. Second, to rule out model dependence, we conducted an additional experiment using Gemini 2.5 Flash-Lite for the simulator while keeping the policy on GPT-4o-mini. The results are shown below:
>
> | Experiment Setup | Simulator Model | Policy Model | $N_{\text{trials}}$ (Lower is better) |
> | :--- | :--- | :--- | :---: |
> | Cross-Model Validation | Gemini 2.5 Flash-Lite | GPT-4o-mini | 19.56 |
>
> This cross-model setup achieved 19.56 trials (still significantly superior to the 28.60 baseline), proving that the efficiency stems from our strategy rather than shared underlying model knowledge.
>
> **Q4:** What steps were taken to confirm that CSX-Sim’s behavior reflects real experimental dynamics rather than learned text correlations?
>
> **A4:** We ground the simulator in the universal scientific principle that structure determines function (lines 83-89)—where lexical patterns in nomenclatures naturally reflect underlying mechanistic similarities (e.g., lattice structures in physics or functional groups in chemistry). Crucially, CSX-Sim transcends surface text correlations by explicitly decomposing hypotheses into interacting components (Figure 3) to model their specific contributions. The ablation study in Table 1 proves that performance significantly degrades without this ComponentExtraction, confirming that the system relies on this fine-grained mechanistic reasoning rather than superficial text-pattern matching.
>
> **Q5:** How sensitive are the results to the choice of LLM (e.g., GPT-4o-mini vs other models) and to changes in the simulator’s noise assumptions?
>
> **A5:** We demonstrate high robustness across model architectures and environmental noise. Regarding LLM choice, we validated the framework by swapping models for both the simulator and the policy.  As detailed in the newly added Appendix K, our method consistently outperforms the baseline regardless of the underlying LLM, and notably achieves even better performance (14.12 trials) with a stronger reasoning model:
>
> | Configuration (Simulator / Policy) | Avg. Trials (Lower is better) |
> | :--- | :--- |
> | Baseline (Pre-Experiment Ranking[1]) | 28.60 |
> | Ours (Gemini 2.5 Flash-Lite / GPT-4o-mini) | 19.56 |
> | Ours (GPT-4o-mini / GPT-4o-mini) | 15.20 |
> | Ours (GPT-4o-mini / Claude 3 Sonnet) | 14.12 |
>
> *Claude 3 Sonnet knowledge cutoff: Aug 2023 (ensuring zero contamination against 2024-2025 data).
>
> This confirms our component-based reasoning is model-agnostic and scales with model capabilities. Regarding noise assumptions, we stress-tested the policy under Simple, Medium, and Complex distortion levels (Section 4.4, Table 3). While efficiency naturally correlates with signal fidelity, CSX-Rank consistently outperforms ablated variants across all regimes (e.g., 32.7 vs. 40.5 trials under Complex noise), demonstrating effective mitigation of misleading signals through structural analysis.
>
> [1] MOOSE-Chem: Large Language Models for Rediscovering Unseen Chemistry Scientific Hypotheses, ICLR 2025.

---

> ### Author Response · Authors · 2025-11-21
> **Reply to Reviewer HGjD - Part 2/2**
>
> **Q6:** Can the authors provide examples where CSX-Rank fails to identify the ground truth or diverges from real trends, and analyze the reasons behind those cases?
>
> **A6:** We thank the reviewer for the suggestion. We identified that failures primarily originate from the model's bias in analyzing complex physicochemical mechanisms. A concrete instance occurred in thermoelectric materials: the ground truth introduced Guanidinium Sulfate to interact with the Potassium ferricyanide / Potassium ferrocyanide redox pair. This addition boosts the Seebeck coefficient ($S_e$) by increasing the reaction entropy difference ($\Delta S_{rc}$), following the thermodynamic relation $S_e = \frac{\Delta E}{\Delta T} = \frac{\Delta S_{rc}}{nF}$. Furthermore, the system involves complex transport dynamics governed by the  Eastman entropy of transfer which drives the thermal diffusion potential ($S_{td}$). However, the ranking policy overlooked these intricate entropic contributions ($\Delta S_{rc}$ and $\hat{S}_i$). Instead, it misinterpreted the sulfate solely as a salting-out agent for  mechanical reinforcement, leading to an underestimation of the hypothesis's rank. We have incorporated this detailed failure analysis into the newly added Appendix L.1.
>
> **Q7:** The framework assumes that hypotheses can be decomposed into functional components and compared through additive similarity. How does it handle cases where scientific mechanisms are emergent or not easily decomposable (e.g., in physics or biology)?
>
> **A7:** Thank you for this insightful comment. We address non-compositional and emergent properties through mathematical gating, cross-disciplinary validation, and rigorous stress-testing:
>
> *   *Mathematical Handling of Emergence:* Our scoring model (Eq. 3) is not merely additive; it crucially incorporates a multiplicative gating term ($\prod_{i \in \mathcal{C}} \mathbf{1}_{s_i > 0}$). This ensures that if a "Critical Point" is missing, the score drops to zero, effectively modeling "all-or-nothing" emergent behaviors. Furthermore, our decomposition (Step 1 in Figure 3 & Appendix A) is performed at the functional/mechanistic level rather than the atomic level. This mirrors how scientists conceptualize hypotheses—treating high-level emergent phenomena (e.g., "ion transport channels") as distinct functional units.
> *   *Generalization to Other Sciences:* This "Functional Decomposition + Critical Gating" logic generalizes well to other experimental sciences where "structure determines property." For instance, in physics, a high-temperature superconductor hypothesis decomposes into functional components: cuprate planes (superconducting mechanism), charge reservoir layers (doping), and interlayer spacing (strain). Through collaboration with experts across experimental physics, chemistry, and biology, we confirmed this abstraction is valid for a broad class of experimental domains, though we acknowledge limitations in purely abstract theoretical physics where such decomposition is less applicable. We have incorporated this applicability analysis into the newly added Appendix L.2.
> *   *Robustness to Non-Linearity:* We empirically validated robustness against non-compositional complexity. We introduced "Cliffs" in our distortion experiments (Section 4.4 & Appendix E) to simulate scenarios where similarity does not map linearly to performance. Our results demonstrate that the system maintains high ranking efficiency even in these non-smooth, highly non-linear landscapes.
>
> ---
>
> We greatly appreciate your comprehensive feedback. We hope that our responses have satisfactorily addressed all your queries. Should you have further questions or suggestions for enhancing our manuscript, we warmly welcome your input.

---

### Official Review · Reviewer_MP1z · 2025-11-01

**Soundness:** 1
**Presentation:** 2
**Contribution:** 1
**Rating:** 0
**Confidence:** 4

**Summary:**

**Goal:** improve ability of AI systems to rank hypotheses in the natural sciences

**Challenge:** running experiments to develop hypothesis ranking strategies is expensive / not always feasible

**Proposed solution:**
1. develop a simulator that models hypothesis "performance" based on similarity to known experiments
2. use this simulator to benchmark an inference strategy for hypothesis ranking ("In Context RL")

**Results:**
1. Simulator outperforms a "matched score" baseline in terms of hypothesis ranking based on correlation with observed hypothesis performance
2. Ranking strategy outperforms randomly selected hypotheses or selection only using base model with no experiment feedback

**Strengths:**

The authors focus on important challenges in automated science and present interesting intuitions for how to generalize knowledge from observed experiments using LLMs based on similarity.

**Weaknesses:**

1. The paper fails to acknowledge or build upon rich prior research in this area, including the fields of active learning and "sim2real" for learning policies from simulators in scientific settings. The paper claims to introduce "experiment-guided ranking" which is more commonly referred to as active learning
2. The simulator is insufficiently validated to support the claim that can be used to meaningfully develop or benchmark hypothesis ranking strategies. Experiments focus on "30 research questions and 124 hypotheses" and assess correlation, which does not provide a basis for robust statements about the utility of this simulator or policies learned upon it. This could also use stronger baselines.
3. Baselines are too weak to make claims about how useful the "in context RL" strategy is. It also doesn't make much sense to refer to this as RL, since the policy is fixed. Overall this experiment could be strengthened substantially: benchmark based on real experiment outcomes, use stronger baselines, and show that the proposed strategy based on decomposing hypotheses and comparing similarity of key components is effective for hypotheses across more settings.

**Questions:**

Overall, there are interesting ideas but the experiments are insufficient to support claims about the value of the proposed simulation and hypothesis ranking strategy. For the simulator, it is not clear that having a better correlation than the matched score baseline is sufficient to show that 1) a policy that performs better with the simulator will be better on real data and 2) that this simulator is good enough to learn a policy that will achieve "sim2real" transfer. For the hypothesis ranking strategy, it would be much stronger to show that this approach achieves better hypothesis ranking that strong baselines across multiple settings with real data, not the simulator.

---

> ### Author Response · Authors · 2025-11-21
> **Reply to Reviewer MP1z  - Part 1/2**
>
> We appreciate your insightful questions and helpful suggestions. For clarity, we have organized your queries along with our responses below:
>
> ---
>
> **Q1:** Lack of acknowledgment of prior work in Active Learning and Sim2Real; claim that "Experiment-Guided Ranking" is commonly referred to as Active Learning.
>
> **A1:** We thank the reviewer for highlighting the connections to Active Learning (AL) and Sim2Real. While our approach leverages iterative feedback [1], it exhibits significant differences from these domains. We have specifically prepared an extensive section in Appendix J.
>
> *   *Task Divergence:* The critical distinction lies in the scope: our framework is universally adaptable for selecting full hypotheses (e.g., choosing between fundamentally different synthesis pathways), whereas AL [2, 4, 5] is limited to focusing on specific parameters inside a fixed hypothesis (e.g., tuning molar ratios [2]). This extends the scope beyond simple parameter tuning.
>
> *   *Training-Free Paradigm for Data Scarcity:* Standard AL (e.g., in nanocatalysis [2] and bio/drug discovery [3, 4]) relies on iterative model retraining. Similarly, Sim2Real methods [8] typically require collecting real-world data to fine-tune policies. In contrast, our In-Context Reinforcement Learning (ICRL) framework remains frozen, enabling efficient reasoning from the very first trial without the high data cost of gradient updates or fine-tuning.
> *   *Optimization-Centric Objective:* AL applications in physics [5] and chemical space [6, 7] primarily target Generalization (minimizing global uncertainty). In contrast, our objective is strictly Optimization (Eq. 6): focusing solely on the path to the optimal hypothesis $h^*$ to minimize trial costs.
> *   *Interpretability:* Instead of opaque scalar acquisition functions (e.g., UCB) common in AL [1, 6], we employ structure-aware reasoning. As detailed in §3.2, our agent prioritizes candidates via component clustering, allowing experts to explicitly audit the decision logic.
>
> [1] Active learning literature survey, 2009.
>
> [2] Active Learning-Driven Discovery of Sub-2 Nm High-Entropy Nanocatalysts for Alkaline Water Splitting, Adv. Funct. Mater. 2025.
>
> [3] Large scale active-learning-guided exploration for in vitro protein production optimization, Nat. Commun. 2020.
>
> [4] Traversing chemical space with active deep learning for low-data drug discovery, Nat. Comput. Sci. 2024.
>
> [5] Active Learning in Physics: From 101, to Progress, and Perspective, 2023.
>
> [6] Less is more: Sampling chemical space with active learning, J. Chem. Phys. 2018.
>
> [7] Chemical Space Exploration with Active Learning and Alchemical Free Energies, J. Chem. Theory Comput. 2022.
>
> [8] Overcoming the Sim-to-Real Gap: Leveraging Simulation to Learn to Explore for Real-World RL, NeurIPS 2024.
>
> **Q2:** Questions regarding the sufficiency of simulator validation to support strategy benchmarking and the utility of learned policies, specifically concerning dataset scale, evaluation metrics, and baseline strength.
>
> **A2:** We thank the reviewer for this thoughtful comment. We demonstrate that the simulator serves as a robust and meaningful testbed for strategy development, evidenced by contamination-free high-value data, strict order-preserving evaluations against SOTA, and demonstrated downstream efficacy in real-world discovery tasks. We provide a detailed elaboration below to further clarify the rigorous foundations of our simulator.
>
> *   *High-Value Data Integrity:* Unlike abundant digital datasets, our benchmark consists of 124 high-cost wet-lab results densely curated from 2024–2025 literature and partner labs, covering diverse domains (e.g., Materials, Physics) as detailed in Table 5. Crucially, this post-dates the model's knowledge cutoff (Oct 2023), ensuring zero data contamination.
> *   *Rigorous Validation Protocol:* We go beyond simple correlation by employing Perfect Consistency Indicator (PCI) and RMSE (Tables 1 & 7) to strictly assess the simulator's ability to preserve relative ordering—which is the fundamental requirement for effective ranking strategies. We further benchmark against the SOTA "Matched Score" [9] (Table 1) to ensure robust performance.
> *   *Demonstrated Real-World Efficacy:* The simulator successfully facilitated the development of CSX-Rank, which achieved a 53% efficiency gain (Table 2) when transferred to the external TOMATO-chem benchmark [9] (comprising real-world scientific problems). Consequently, the significant performance gains of the simulator-derived policy on real-world tasks strongly validate its practical effectiveness for benchmarking and strategy development.
>
> [9] MOOSE-Chem: Large Language Models for Rediscovering Unseen Chemistry Scientific Hypotheses, ICLR 2025.

---

> > ### Author Response · Authors · 2025-11-21
> > **Reply to Reviewer MP1z - Part 2/2**
> >
> > **Q3:** Concerns regarding baseline strength; skepticism about referring to the strategy as "RL" given that the policy is fixed, and the need for real-world benchmarking.
> >
> > **A3:**  We thank the reviewer for the constructive feedback. Regarding "In-Context RL", we strictly align with the established paradigm in Lee et al. [10], which demonstrates that transformers with fixed parameters perform reinforcement learning by adapting via history accumulation (context) rather than weight updates.
> >
> > Regarding strategy baselines, in addition to Yang et al. [9], we added a comparison with Si et al. [11] on the TOMATO-chem benchmark. It is important to note that the TOMATO-chem benchmark [9] is constructed from 51 real-world scientific discovery problems extracted from top-tier literature. Our superior performance on this benchmark directly demonstrates the effectiveness of our decomposition-based strategy in realistic scientific settings. As shown below, the baseline methods require significantly more trials than ours:
> >
> > | Method | Avg. Trials (Lower is better) |
> > | :--- | :--- |
> > | Si et al. [11] | 24.68 |
> > | Yang et al. [9] | 28.61 |
> > | **CSX-Rank (Ours)** | **15.20** |
> >
> > [10] Supervised Pretraining Can Learn In-Context Reinforcement Learning, NeurIPS 2023.
> >
> > [11] Can LLMs Generate Novel Research Ideas? arXiv 2024.
> >
> > ---
> > We greatly appreciate your comprehensive feedback. We hope that our responses have satisfactorily addressed all your queries. Should you have further questions or suggestions for enhancing our manuscript, we warmly welcome your input.

---

### Author Response · Authors · 2025-12-02
**Thank All Reviewers**

To all reviewers:

We would like to thank all reviewers for their thoughtful insights and valuable comments.

We summarize the contributions of this paper below:

1. We introduce and formalize the new task of "Experiment-Guided Ranking". It is the first to leverage experimental feedback compared to the previous hypothesis ranking methods. Distinct from Active Learning—which is limited to tuning specific parameters within a fixed hypothesis—our task focuses on ranking full and complete hypotheses, which aligns much closer to the real experiment setting.
2. We highlight a key bottleneck of "Experiment-Guided Ranking" in the natural sciences: the lack of scalable access to wet-lab experimental feedback. To address this, we propose the use of simulators and release a curated dataset of 124 scientific hypotheses with annotated performance collected from the literature and real wet-lab experiments.
3. We introduce three conceptual foundations characterizing the latent performance landscape of scientific hypotheses. We mathematically formalize this simulation process and construct a high-fidelity simulator that approximates real wet-lab outcomes by modeling performance as a function of hypothesis similarity and systematic distortion.
4. We introduce a clustering-based agentic ranking policy implemented within an in-context reinforcement learning framework. By leveraging functional decomposition to attribute experimental feedback to the marginal utility of components, our method conceptually transforms the discovery problem from an exponential combinatorial search into a linear component optimization. This allows the agent to generalize efficiently from limited feedback, empirically outperforming both pre-experiment baselines and ablation variants.

We are excited that you recognized our contributions. We quote correspondingly as below:

1. "The authors focus on important challenges in automated science..." [Reviewer MP1z]; "This paper studies the important problem of automated hypothesis discovery when real-world feedback is limited." [Reviewer 2keb]
2. "The proposed benchmark and hypotheses may be of use for future studies in the field." [Reviewer 2keb]; "Demonstrates high correlation with 124 real experiments." [Reviewer HGjD]
3. "Strong, well-motivated design of the simulator (CSX-Sim)..." [Reviewer HGjD]; "Present interesting intuitions for how to generalize knowledge from observed experiments using LLMs based on similarity." [Reviewer MP1z]
4. "Strong... design of the... in-context RL policy (CSX-Rank)." [Reviewer HGjD]; "Clear efficiency gains over baselines." [Reviewer HGjD]; "The framework is shown to outperform other baselines." [Reviewer 2keb]

We are grateful that you also found that:

* "The paper is generally well-written and well-structured, with clear figures and examples." [Reviewer HGjD]
* "The paper is clearly structured." [Reviewer 2keb]

We also appreciate many helpful suggestions, based on which we have improved our manuscript. The main changes are:

* We added Appendix J to explicitly differentiate our framework from Active Learning and Sim2Real. We highlighted the Task Divergence (selecting full hypotheses rather than parameter tuning) and our Training-Free, Optimization-Centric paradigm.
* We added Appendix K containing a sensitivity analysis, proving our method's robustness across different LLMs (Gemini, Claude 3 Sonnet) and varying simulator noise levels.
* We added Appendix L to analyze failure modes (e.g., complex thermodynamic mechanisms) and discussed how our method handles emergent properties via multiplicative gating.
* We added a stronger baseline comparison (Si et al., 2024) on the TOMATO-chem benchmark to further validate efficiency.
* We expanded the discussion on the generalizability of our "Functional Decomposition" logic to Physics, Materials Science, and Biology (Appendix L.2 and Appendix M), supported by empirical validation across these domains (Table 5).
* Added Theoretical Formalization (Section 4.3 & Appendix M): Formally demonstrated that functional decomposition transforms hypothesis discovery from an exponential combinatorial search into a linear component optimization, mathematically grounding our empirical efficiency gains.

We would again like to thank all reviewers for their time and effort, and we hope that our changes adequately address all concerns. We are glad to have further discussions and take additional suggestions to help improve our manuscript.

Sincerely,

Authors

---

### Author Response · Authors · 2025-12-03
**Author Final Remarks**

Dear AC and Reviewers,


We sincerely thank the AC and all reviewers for their thoughtful engagement with our submission. The contributions of the paper are outlined in the “Thank All Reviewers” comment, and we are pleased to note that each contribution has been endorsed by at least one reviewer, with the relevant comments from the reviewers carefully selected and included in the comment.


From the authors' perspective, we believe we have thoroughly addressed all the concerns raised by the reviewers. We have responded to each question, engaged in detailed discussions regarding the distinction between our work and Active Learning, and incorporated numerous additional experiments to reinforce our arguments. Furthermore, we have comprehensively resolved all feedback from a previous NeurIPS submission cycle, ensuring that the manuscript has evolved into a highly robust and mature contribution.


Notably, based on constructive feedback, we have significantly strengthened the manuscript during the rebuttal by:
1. *Solidifying the Theory:* Adding a formal proof of search complexity reduction (Section 4.3 & Appendix M) to ground our empirical gains.


2. *Verifying Robustness:* Conducting sensitivity analyses across different model families (Gemini, Claude 3 Sonnet) and noise levels to rule out circular logic.


3. *Elaborating on Generalizability:* Expanding the discussion on the generalizability of our "Functional Decomposition" logic to Physics, Materials Science, and Biology (Appendix L.2 and Appendix M), supported by empirical validation across these domains (Table 5).




Thanks to the reviewers' constructive feedback, and with the new experiments conducted during the rebuttal, every claim in the paper is now supported by robust empirical evidence and mathematical grounding.


We firmly believe that our work makes a meaningful contribution to the community. We are also deeply grateful for the kind words from the reviewers, noting that our work addresses "important challenges in automated science" [Reviewer MP1z], studies an "important problem... when real-world feedback is limited" [Reviewer 2keb], and demonstrates "strong, well-motivated design... and clear efficiency gains over baselines" [Reviewer HGjD].


Sincerely,


Authors

---

### Meta-Review · Area_Chair_Fdhv · 2026-01-02

**Summary:**

This paper proposes an LLM-based simulator of scientific experiments as well as an LLM-based in-context RL policy to rank scientific hypotheses. Though a promising idea, there are several limitations with the paper as it is:

1. The motivations and methods of this paper are very related to numerous established subfields within the ML/AI community: sim2real transfer, active learning, sequential experimental design, Bayesian optimization, etc. While the authors note differences from their proposed setting/approach, the complete lack of discussion or comparison with these fields makes it challenging to assess the utility, novelty, or significance of the proposed method relative to these other approaches, which have established use and efficacy in scientific settings.
2. The scope of the method is both too narrow and too large. The authors primarily focus on chemistry experiments, with some physics and biology experiments. The authors claim that their method is suitable for ranking all scientific hypotheses. However, their method assumes that hypotheses can be decomposed into functional components, and many fields may have competing definitions of what a hypothesis is. An evaluation of 124 hypotheses is far too limited to make sweeping claims about a general-purpose simulator.

I suggest a major revision of the paper that (A) discusses and/or compares techniques with related subfields and (B) limits the scope to a single setting (e.g., a subdiscipline of chemistry) to make the evaluation more comprehensive and to establish trust. As it is, the current version of the paper is not suitable for publication at ICLR.

**Reviewer Concerns:**

The reviewer concerns addressed by the rebuttal include:

* Confirmation that the simulator reflects real-world experimental dynamics
* The testing framework includes a knowledge cutoff to prevent testing leakage.

The primary concerns around (1) scope/generality of the method and (2) relation to other subfields were addressed by the authors, but in my opinion, the authors’ remarks were insufficient to quell these concerns. I don’t believe that the authors’ remarks about scope and relation to subfields would have been resolved with more discussion. As stated in the metareview, I believe the only way to resolve these concerns would be to conduct additional experiments and engage in further discussion, which would constitute a major revision to the paper.

**Reviewer Scores:**

Because none of the reviewers engaged at all, I do not think that the reviewers would have engaged with this paper and changed their scores even if the discussion period had not been cut short. I acknowledge that this outcome is unfortunate and a poor reflection on the reviewers, but the outcome is no more unfair than it would have been under a normal reviewing period.

---

### Decision · Program_Chairs · 2026-01-26

Reject